# Seasonal switching of integrated leaf senescence controls in an evergreen perennial *Arabidopsis*

Genki Yumoto [1] ✉, Haruki Nishio [1,2], Tomoaki Muranaka [1,3], Jiro Sugisaka[1], Mie N. Honjo [1] & Hiroshi Kudoh [1] ✉

Evergreeness is a substantial strategy for temperate and boreal plants and is as common as deciduousness. However, whether evergreen plants switch foliage functions between seasons remains unknown. We conduct an *in natura* study of leaf senescence control in the evergreen perennial, *Arabidopsis halleri*. A four-year census of leaf longevity of 102 biweekly cohorts allows us to identify growth season (GS) and overwintering (OW) cohorts characterised by short and extended longevity, respectively, and to recognise three distinct periods in foliage functions, i.e., the growth, overwintering, and reproductive seasons. Photoperiods during leaf expansion separate the GS and OW cohorts, providing primal control of leaf senescence depending on the season, with leaf senescence being shut down during winter. Phenotypic and transcriptomic responses in field experiments indicate that shade-induced and reproductive-sink-triggered senescence are active during the growth and reproductive seasons, respectively. These secondary controls of leaf senescence cause desynchronised and synchronised leaf senescence during growth and reproduction, respectively. Conclusively, seasonal switching of leaf senescence optimises resource production, storage, and translocation for the season, making the evergreen strategy adaptively relevant.

In regions with significant seasonal temperature variations such as temperate and higher latitudes, plants experience optimal and suboptimal periods for growth, that is, growing and overwintering seasons. During the growing season, plants alter leaf characteristics to maximise photosynthetic production at the whole-plant level, while minimising storage loss becomes a primary requirement during winter[1]. Deciduousness, that is plants effectively switching between growing and overwintering seasons by accelerating leaf senescence in autumn and becoming dormant during winter, is one of the most efficient strategies of effectively coping with this situation[2]. However, it is unclear whether evergreen plants undergo a seasonal switch similar to that of deciduous plants. Although evergreen plants have leaves throughout the year, the control of leaf senescence may differ

between the growing and overwintering season, and distinct controls may result in differences in the spatial arrangement and age composition of the leaves (age structure).

The most immediate cause of leaf senescence is the developmental aging of individual leaves, known as age-dependent senescence[3,4]. The efficiency of photosynthetic assimilation decreases with leaf age after leaf expansion. When this efficiency declines to a particular level, the senescence program is activated to systematically degrade chloroplasts and catabolise cellular molecules to translocate resources that can be used for new organ formation[3,5]. The molecular mechanisms of age-dependent senescence have been extensively studied, and key transcription factors, such as *ORESARA 1* (*ORE1*) and *NAC-LIKE ACTIVATED BY AP3/PI* (*NAP*), positively regulate leaf

[1]Center for Ecological Research, Kyoto University, Hirano 2-509-3, Otsu 520-2113, Japan. [2]Data Science and AI Innovation Research Promotion Center, Shiga University, Banba 1-1-1, Hikone 522-8522, Japan. [3]Graduate School of Bioagricultural Sciences, Nagoya University, Furo-cho, Chikusa-ku, Nagoya 464-0814, Japan. ✉e-mail: gyumoto@ecology.kyoto-u.ac.jp; kudoh@ecology.kyoto-u.ac.jp

senescence by activating genes encoding chlorophyll catabolic enzymes, such as *NON-YELLOW COLORING 1* (*NYC1*), *NONYELLOWING 1* (*NYE1*), and *NYE2*[6]. Leaf senescence is further accelerated by self-shading[7,8]. The mechanism of shade-induced senescence has been investigated[9], often using dark-induced senescence as an extreme model[10,11]. PHYTOCHROME INTERACTING FACTOR 4 (PIF4) and PIF5 proteins are released from PHYTOCHROME B (phyB) in the dark, where phyB becomes inactive, and then binds to the promoters of *ORE1*, *ABA INSENSITIVE 5* (*ABI5*), *ENHANCED EM LEVEL* (*EEL*), and *ETHYLENE-INSENSITIVE 3* (*EIN3*) to, directly and indirectly, upregulate *ORE1*[11,12]. Theoretical studies support the idea that the active senescence of old and shaded leaves for replacement by new leaves optimises growth at the whole-plant level regarding carbon and nutrient economies[13–15].

High nitrogen and phosphorus demand for seed production triggers leaf senescence, even in young mature leaves, and activates the translocation of nutrients to the reproductive sink (referred to as reproductive-sink-triggered senescence)[4]. In soybeans, removal of the reproductive sink delays leaf senescence[16,17]. Transcriptome analysis in soybean showed that storage-protein genes were upregulated in sink-limited plants, whereas senescence-related transcription factors (NACs and WRKYs) were upregulated in control leaves[18]. Leaf longevity responses to manipulation of the reproductive sink are weak in *Arabidopsis thaliana*[19] and the molecular mechanisms of reproductive sink-triggered senescence have been poorly explored in this species. However, a recent study suggested that bolting-associated genes in *A. thaliana* include senescence-related NACs and WRKYs[20]. The mechanisms of nitrogen and phosphorus translocation during leaf senescence have been analysed in starvation-triggered senescence, and the involvement of macromolecule (proteins, nucleic acids, and lipids) degradation- and autophagy-related genes has been suggested[21,22].

These indicate that leaf senescence is a highly complex process triggered by multiple internal and external cues[6,23,24]. Although the control mechanisms of leaf senescence responding to each cue have been extensively studied, there is a deficit in our understanding of how these multiple mechanisms coordinate to maximise plant fitness in natural environments. For instance, we do not know how leaf senescence is controlled during winter or its consequences to the entire plant. Studying evergreen plants under natural conditions provides critical information regarding the mechanisms controlling leaf senescence during contrasting seasons.

Herein, we investigated the potential existence of a clear switch in the control of leaf senescence between the growing and overwintering season in the evergreen perennial, *Arabidopsis halleri* subsp. *gemmifera*.

## Results

### Leaf emergence and longevity show clear seasonality
We conducted a four-year *in natura* study of leaf longevity and foliage dynamics in a natural habitat of *A. halleri* in central Japan (Supplementary Fig. 1a, b)[25]. Thirty new plants were allocated each October (Supplementary Fig. 1c), except for the fourth year where we used the same individuals utilised in the third year. Newly emerged leaves were counted every two weeks (biweekly cohorts) and distinguished using coloured strings (Supplementary Fig. 1d). The survival of tagged leaves was recorded weekly to determine leaf longevity. Leaf length was measured biweekly to determine leaf growth rate in terms of length (mm/d), leaf growth period (d), and mature leaf length (mm).

Temperature and photoperiod showed seasonal oscillations, with photoperiods being advanced by approximately 1.5 months, indicating that they are different cues in terms of seasonal timing (Fig. 1a). The seasonality of leaf emergence, longevity, and growth rate was evident in the 102 biweekly cohorts over the four years (Fig. 1b–d). Leaf emergence was high during the growing season (March–September)

and low during the overwintering season (October–February, Fig. 1b). Mean leaf longevity of the cohorts ranged between 26.7 and 176.2 d, with an average of 83.5 d (Supplementary Data 1). The maximum longevity of all 3334 tagged leaves was 308 d, recorded for a leaf that emerged on 5 November 2019. Cohorts that emerged between March and September had shorter longevity and higher leaf growth rates, whereas those that emerged between October and February had extended longevity and lower leaf growth rates (Fig. 1c, d). For the cohorts that emerged between March and September, the growth rate was relatively lower in the June and July cohorts (and those that emerged in neighbouring periods, depending on the year) than in the remaining cohorts (Fig. 1d). As the growth period of these leaves extends from June to September, the high-temperature regime may explain the suppressed growth in summer.

### Leaf longevity and growth define the growth season (GS) and overwintering (OW) cohorts
k-means clustering using leaf longevity represented by days to 50% survival ($L_{50}$) and leaf growth rate separated the leaf cohorts into two groups (Fig. 1e; the optimal number of groups was estimated as two by the silhouette method[26], Supplementary Fig. 2a). Cohorts that emerged during March–September and October–February were designated as growth season (GS) and overwintering (OW) cohorts, respectively, except for some late February cohorts that were classified as GS cohorts, depending on the year (Fig. 1e). $L_{50}$ of the GS cohorts averaged 59.8 d and ranged from 26.7 to 92.4 d, while $L_{50}$ of the OW cohorts averaged 129.7 d and ranged from 81.4 to 176.2 d.

To elucidate the environmental determinants separating the GS and OW cohorts, we conducted decision tree analysis (DTA) using photoperiod, temperature, and solar radiation data (averaged over 33 d after leaf emergence, which was the median number of days until all measured leaves reached 90% of their mature leaf length; the values were lower than $L_{50}$ for most cohorts). DTA found that the GS and OW cohorts were separated by photoperiod alone without error at a threshold of 11.4 h. As expected from DTA, both $L_{50}$ and leaf growth rate showed a clear GS-OW separation by photoperiod rather than temperature (Fig. 1f, g, Supplementary Fig. 2b, c).

### Dynamics of individual leaves result in seasonal age structures, optimising production and storage
The combination of seasonal patterns of leaf emergence and senescence resulted in dynamic changes in the leaf age structure at the whole-plant level (Fig. 1h, i, Supplementary Fig. 3). The June–September period was characterised by many extant leaves of 2 months and younger (Fig. 1h, Supplementary Fig. 3) and many withered leaves aged 1–4 months (Fig. 1i), representing a foliage structure for active photosynthetic production with high leaf turnover. The number of extant leaves decreased from October to February, whereas the age structures gradually developed to include older leaves (as old as 8 months) by early May (Fig. 1h, Supplementary Fig. 3), representing a foliage structure optimising the resource storage function prior to reproduction. The number of withered leaves remained low during winter, and many overwintered leaves withered synchronously in May during fruit and seed maturation (Fig. 1i).

### GS and OW cohorts show successive and synchronised senescence
Placing the emergence date and $L_{50}$ of the 102 cohorts on the calendar revealed that the GS cohorts senesced sequentially, depending on the days after emergence, suggesting that senescence control is primarily controlled by the age of individual leaves (Fig. 2a). In contrast, the OW cohorts showed extended longevity and synchronised $L_{50}$ timings in terms of calendar days, corresponding to the onset of flowering from March to May. This suggests that the primary control of senescence depends on the reproductive schedule at the whole-plant level

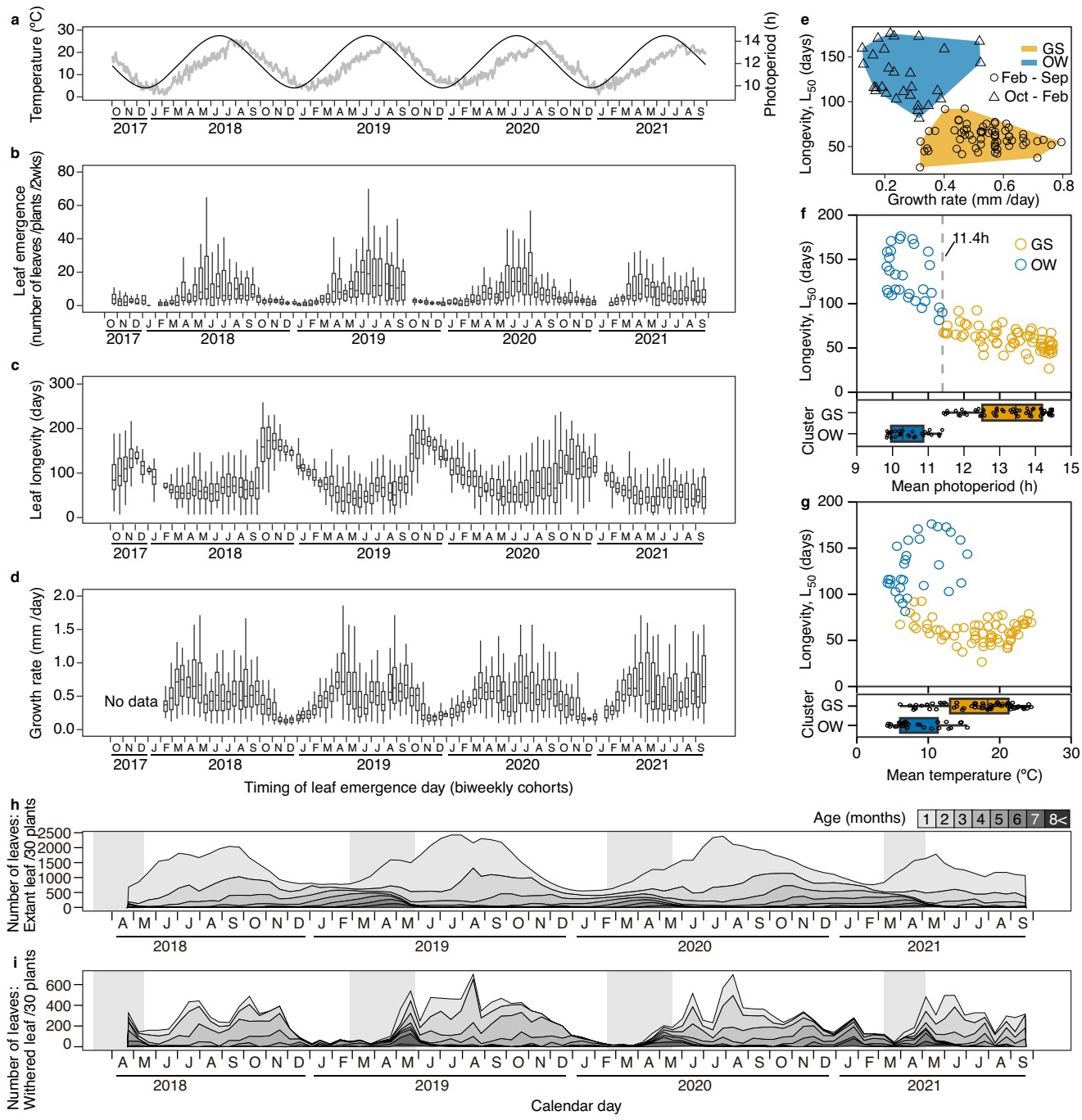

**Fig. 1 | Four-year biweekly cohort analysis of seasonal leaf dynamics of *Arabidopsis halleri* in a natural population.** Seasonal changes in photoperiod (black) and average daily temperature (grey) at the study site (**a**), number of newly emerged leaves per plant over 2 weeks (**b**), leaf longevity (**c**), and leaf growth rate of the biweekly cohort (**d**). **e** *k*-means clustering (*k* = 2) separating the GS and OW cohorts (orange and blue shading, respectively) using $L_{50}$ and leaf growth rate. $L_{50}$ plotted against photoperiod (**f**) and temperature (**g**) experienced by leaf cohorts (averaged over 33 d after leaf emergence). The grey dotted line in **f** indicates threshold obtained from the decision tree analysis to classify GS and OW cohorts. The Box plots in **f**, **g** below show the differences between the GS and OW cohorts in the

respective environmental factors (**f, g**). In the box plots (**b–d, f, g**), 25%, 50% (=$L_{50}$ in **c**), and 75% of the data were indicated by hinges and centre lines. The whiskers extend from the hinges to the highest and lowest values, which are within 1.5× of the box lengths. Seasonal changes in leaf age structure. Total number and leaf age compositions of extant (**h**) and withered (**i**) leaves. Leaves are pooled and adjusted to the total number of leaves of the 30 plants. Shaded square areas in **h, i** represent reproductive periods (from onset of bolting to end of flowering). Leaf age classes are indicated by grey scales (scale bar at top). Leaf longevity and growth rate were measured using 3334 leaves (102 cohorts). Source data are provided as a Source Data file, and the number of replicated leaves for each of the cohorts is provided.

(Fig. 2a). Reversion from reproductive to vegetative growth occurred in May–June after synchronised senescence of the OW cohorts (Fig. 2a).

The $L_{50}$ and survival plots showed distinct patterns in the GS and OW cohorts (Fig. 2b). For each of the 102 biweekly cohorts, we fitted an

equation, *survival rate* = exp $(-(x/\beta)^{\alpha})$, where *x* is the number of days after leaf emergence, and $\alpha$ and $\beta$ are shape and scale parameters, respectively (Supplementary Fig. 4)[27]. In this equation, $\alpha$ is the approximate synchronisation of leaf senescence within a cohort. The OW cohorts initially showed low mortality and later highly

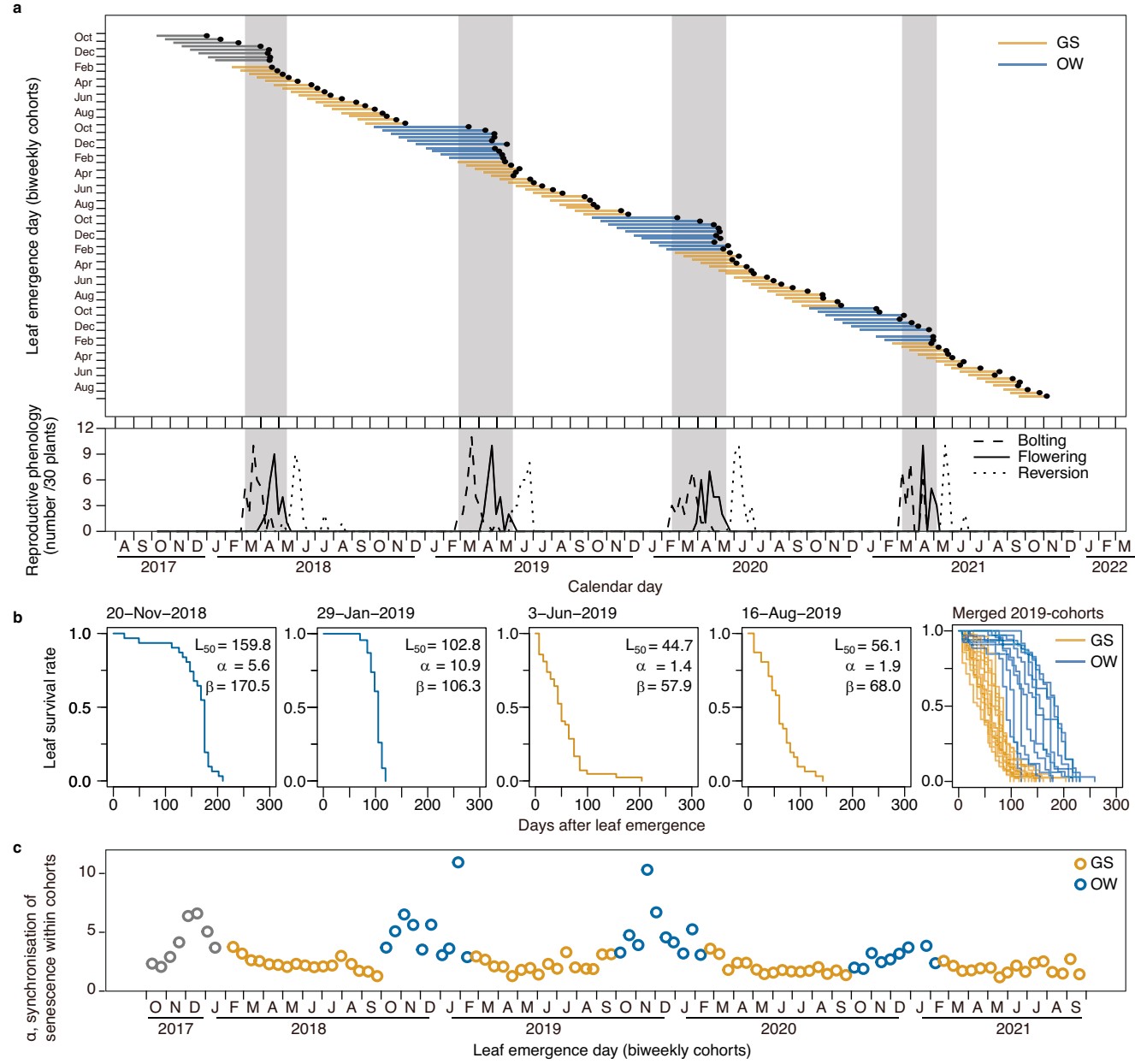

**Fig. 2 | Seasonal dependence of senescence synchronisation of GS (orange) and OW (blue) cohorts. a** Synchronisation of $L_{50}$ with calendar day and reproductive phenology. Leaf longevity (coloured lines) and $L_{50}$ times (black dots) are shown. Shaded square areas correspond to the reproductive period in the diagram below (from onset of bolting to end of flowering). **b** Kaplan–Meier survival plots of representative cohorts. $L_{50}$, shape ($\alpha$), and scale ($\beta$) parameters estimated from the survival curve fitting are given. **c** Seasonal change in $\alpha$, which represents the degree of senescence synchronisation within cohorts. Source data are provided as a Source Data file.

synchronised mortality (higher $\alpha$), as exemplified by the 20 November 2018 and 29 January 2019 cohorts, whereas the GS cohorts showed almost constant mortality ($\alpha$ close to one), as exemplified by the 3 June 2019 and 16 August 2019 cohorts (Fig. 2b). $\alpha$ showed seasonality that peaked in winter (Fig. 2c), suggesting that the synchronisation of leaf senescence in OW cohorts occurred between cohorts and leaves within cohorts. Although the GS cohorts had short leaf longevity on average, low $\alpha$ of the GS cohorts indicated that the timing of senescence was desynchronised between the leaves within the cohorts, which could not be explained by the age of individual leaves.

### Self-shading induces senescence in the growing season
Self-shading may introduce a secondary regulation that explains the desynchronisation of leaf senescence within the GS cohorts, and the extended leaf longevity of the OW cohorts may represent a shutdown of the senescence response. We then experimentally shaded and exposed leaves of the typical GS and OW cohorts that emerged in early July and late January, respectively (referred to as GS and OW self-shading experiments, respectively) (Fig. 3a). In the GS self-shading experiment, accelerated senescence (11.7 d advance in $L_{50}$) was observed in the self-shading treatment relative to that in the exposure (Fig. 3b, Supplementary Data 2). In contrast, self-shading did not induce senescence in the OW self-shading experiment (Fig. 3c, Supplementary Data 2), suggesting that shade-induced senescence is active during growing season, but is deactivated during winter. We performed time-series transcriptome analysis during the GS and OW self-shading experiments at 0, 2, 4, 6, and 8 and 0, 4, 8, and 12 weeks after the start of the treatments, respectively (Fig. 3b, c). At

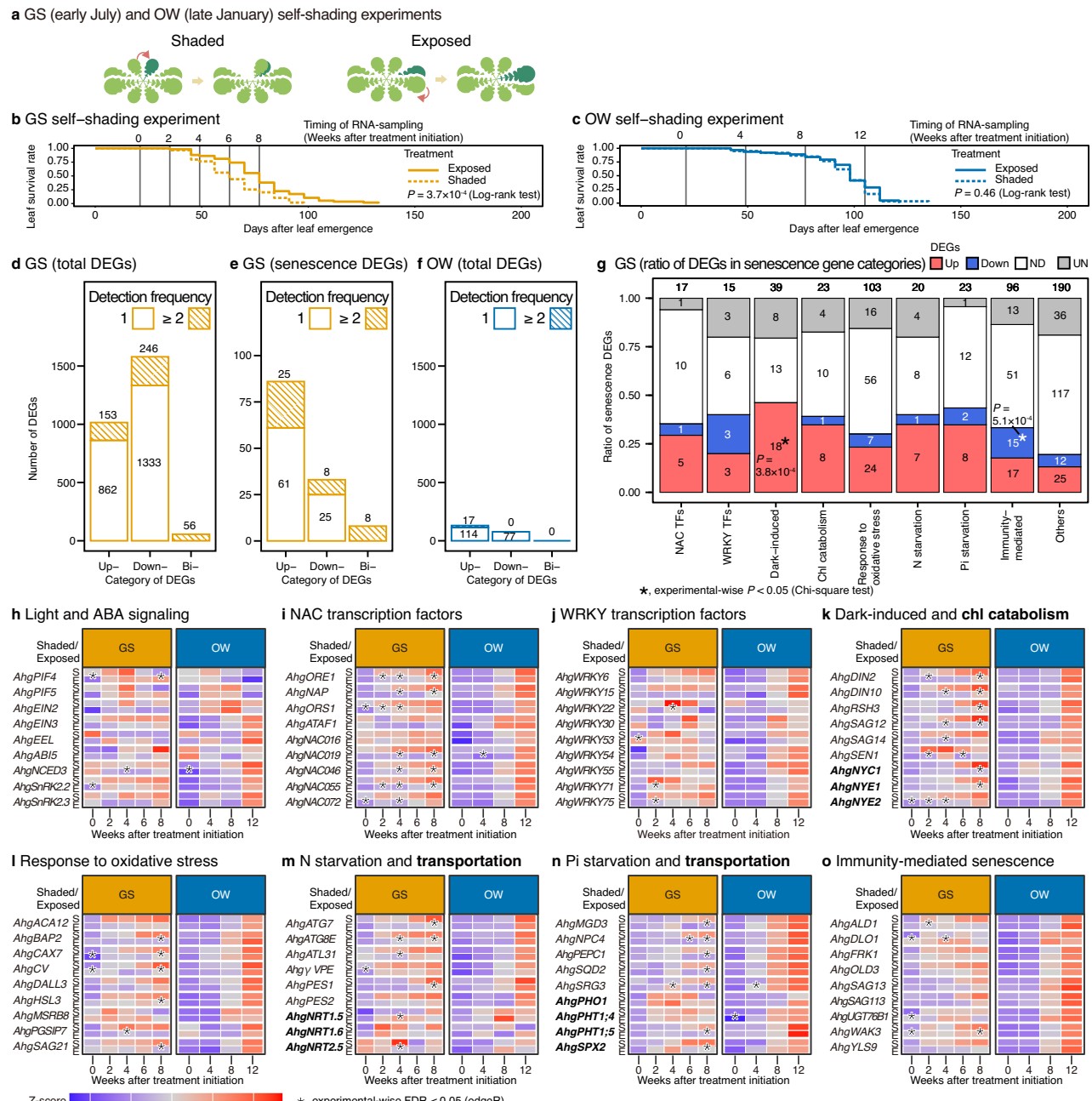

**Fig. 3 | Season-specific sensitivity of leaf senescence to shading in the GS and OW self-shading experiments. a** Treatments of self-shading experiments with representative GS and OW cohorts. A thread was used to move the upper leaf to keep the target leaf (dark green) shaded or exposed (arrow). Responses in leaf longevity in the GS (**b**) and OW (**c**) self-shading experiments. Number of total or senescence-related upregulated, downregulated, and bidirectionally-regulated DEGs in the GS (total, **d**; senescence-related, **e**) and OW (total, **f**) self-shading experiments. We designated higher expression in the senescence-enhancing treatment (i.e., shaded) as upregulation. Upregulated and downregulated DEGs include those significantly upregulated once or at more time points, and vice versa, respectively. The bidirectionally regulated DEGs are those significantly regulated to different directions at different times. **g** Ratio of senescence-related DEGs (figures = number of genes) for

nine categories (total 526 counts for the 432 genes, allowing multiple counts of single genes in different categories). ND and UN represent genes that were not DEGs and unexpressed, respectively. Heat maps showing the time-series of gene expression under the shaded (S) and exposed (E) treatments in the GS and OW cohorts. Results are shown for nine genes related to light and ABA signalling (**h**), NAC transcription factors (**i**), WRKY transcription factors (**j**), dark-induced-senescence (**k**) and chlorophyll catabolism (**k**, bold), response to oxidative stress (**l**), N starvation (**m**) and translocation (**m**, bold), phosphate (Pi) starvation (**n**) and translocation (**n**, bold), and immunity-mediated senescence (**o**). Data are presented as z-scores calculated for each gene across all combinations of cohorts, treatments, and time points. All statistics are two-sided. The multiplicity of the tests is corrected by the Bonferroni method (**g–o**). Source data are provided as a source data file.

0 weeks, samples were collected 1 h after treatment. We then detected the differentially expressed genes (DEGs) between treatments by setting the experiment-wise false discovery rate (FDR) = 0.05 (Fig. 3d–o). We defined upregulated and downregulated genes in senescence-enhanced (shaded) treatments compared to senescence-delayed

(exposed) treatments. Most significantly upregulated and downregulated genes were detected once or multiple times in the same direction, and we treated these as upregulated and downregulated DEGs (Fig. 3d–f). Bidirectional DEGs, which were detected as both upregulated and downregulated, depending on the sampling date,

were rare, and were excluded from the subsequent DEGs analysis (Fig. 3d–f).

The close relatedness of *A. halleri* and *A. thaliana* made it possible to identify pairs of homologous genes based on sequence similarity (represented by the addition of *Ahg* to the gene names of *A. halleri* subsp. *gemmifera*) and we defined 432 senescence-related genes in *A. halleri* based on gene ontology (GO) terms and a literature survey (Supplementary Data 3). In the GS self-shading experiment, we detected 1015 upregulated and 1579 downregulated DEGs (Fig. 3d, Supplementary Data 4), including 86 and 33 senescence-related genes, respectively (Fig. 3e). Although we found a higher number of downregulated DEGs overall, more senescence-related genes were upregulated (Fig. 3d, e). In the OW self-shading experiment, we detected 131 and 77 genes in total, and six and three senescence-related genes, as upregulated and downregulated DEGs, respectively (Fig. 3f, Supplementary Data 3, Supplementary Data 4). The smaller number of DEGs in the OW cohort indicated that the response of gene expression to shading was minimal in winter (Fig. 3f, Supplementary Data 4). The number of enriched GOs (FDR < 0.05) was 65/108 (upregulated/downregulated DEGs) and 16/0 for the GS and OW self-shading experiments, respectively (Supplementary Data 5).

Principal component analysis (PCA) of all samples using all the expressed genes (19,292 genes) showed distinct positioning in the PCA space for samples of shaded leaves collected at 6 and 8 weeks after treatment initiation from other samples in the GS self-shading experiment (Supplementary Fig. 5a). In contrast, the trajectories of the shaded and exposed samples overlapped in the PCA space along the time course in the OW self-shading experiments (Supplementary Fig. 5b). The PCA results suggest transcriptome-level shading-sensitivity and insensitivity in the GS and OW cohorts, respectively.

Based on GOs and literature sources, we classified the 432 senescence-related genes into nine categories, allowing multiple assignments of a gene into different categories (526 total counts for the 432 genes, Supplementary Data 3). Nearly half of the dark-induced senescence genes were detected as upregulated DEGs in the GS self-shading experiment (Fig. 3g), and 13 of these were listed as dark-induced senescence genes in a previous study on *A. thaliana*[28]. In addition, genes classified as being involved in chlorophyll catabolism and nitrogen and phosphate (Pi) starvation showed a relatively high proportion of upregulated DEGs (Fig. 3g).

The time-series expression patterns of senescence-related and selected genes are shown in heat maps (Fig. 3h–o). Among the homologous genes involved in shade-induced senescence that is located upstream of *ORE1* in the signalling pathway, *AhgPIF4* and *AhgPIF5* did not show consistent transcriptional responses (Fig. 3h), illustrating their responses at the protein level[12]. Homologues of the three transcription factors linking PIFs and *ORE1* upregulation, i.e., *AhgABI5*, *AhgEEL*, and *AhgEIN3*, tended to be upregulated in the GS shade treatment, but the differences were not statistically significant (Fig. 3h). For the other selected genes, the sensitivity and insensitivity of the GS and OW cohorts to the shading/exposure treatments, respectively, were also evident at the gene expression level (Fig. 3 i–o). In response to the self-shading of the GS cohort, the upregulated DEGs included many of the NAC transcription factors, such as *AhgORE1*, *AhgNAP*, *ORESARA1 SISTER 1* (*AhgORS1*), *ARABIDOPSIS NAC DOMAIN CONTAINING PROTEIN 19* (*AhgNAC019*), *AhgNAC046*, *AhgNAC055*, and *AhgNAC072* (Fig. 3i), and *WRKYs* (Fig. 3j; in particular, *AhgWRKY22* and *AhgWRKY71* homologues are known transcription factors induced by dark treatment[29,30]). Upregulated DEGs response to shading also included dark-induced senescence and chlorophyll catabolism genes, such as *DARK INDUCIBLE 2* (*AhgDIN2*), *AhgDIN10*, *RELA/SPOT HOMOLOG 3* (*AhgRSH3*), *SENESCENCE-ASSOCIATED GENE 12* (*AhgSAG12*), *AhgSAG14*, *SENESCENCE 1* (*AhgSEN1*), *AhgNYC1*, *AhgNYE1*, and *AhgNYE2* (Fig. 3k), and nitrogen and Pi starvation-induced and transport genes, including *AUTOPHAGY 7* (*AhgATG7*), *AhgATG8E*, *ARABIDOPSIS*

*TOXICOS EN LEVADURA 31* (*AhgATL31*), *PHYTYL ESTER SYNTHASE 1* (*AhgPES1*), *NITRATE TRANSPORTER 1.5* (*AhgNRT1.5*), *AhgNRT2.5*, *MONOGALACTOSYL DIACYLGLYCEROL SYNTHASE 3* (*AhgMGD3*), *NON-SPECIFIC PHOSPHOLIPASE C4* (*AhgNPC4*), *PHOSPHOETHANOLAMINE /PHOSPHOCHOLINE PHOSPHATASE 1* (*AhgPEPC1*), *SULFOQUINOVOSYL-DIACYLGLYCEROL 2* (*AhgSQD2*), *AhgSRG3*, *PHOSPHATE TRANSPORTER 1;5* (*AhgPHT1;5*), and *SPX DOMAIN GENE 2* (*AhgSPX2*) (Fig. 3m, n).

The upregulation of many DEGs was observed 4 weeks after treatment initiation, when the survival plot of the shaded and exposed leaves started to diverge (Fig. 3b, i–o). Active nitrate recycling was suggested by DEGs that are homologous to autophagy-related genes, such as *AhgATG7* and *AhgATG8E*, and nitrogen transporters, such as *AhgNRT1.5* and *AhgNRT2.5* (Fig. 3m). Many oxidative stress response genes were detected as upregulated DEGs in the GS experiment (Fig. 3l), suggesting that reactive oxygen species (ROS) signalling is involved in the accelerated senescence of shaded leaves in the GS cohort. A few immunity-mediated senescence-related genes responded to the treatment in the GS cohort (Fig. 3o). In the OW cohort, few senescence-related genes were detected as DEGs, suggesting that senescence became insensitive to self-shading during winter at the gene regulation level, although they were strongly expressed at week 12 when both shaded and exposed leaves senesced simultaneously (Fig. 3h–o).

## Strong sink demand causes synchronised senescence of the OW cohorts

As a strong synchronisation of senescence was observed between and within the OW cohorts, we expected that seasonal differences in sink demand might induce another type of leaf senescence control. Accordingly, we experimentally removed the sink (new leaves) for the GS cohort at the end of July (GS sink-removal experiment, Fig. 4a) and flowering stalks for the OW cohort at the beginning of February (OW sink-removal experiment, Fig. 4a). We removed the strongest sink depending on the season; the formation of new leaves is most active in summer (Fig. 1b), while the formation of reproductive organs occurs once a year synchronously with the senescence of OW cohorts (Fig. 2a). Sink-removal resulted in the extension of $L_{50}$ by 9.7 and 27.1 d for the GS and OW cohorts, respectively, indicating stronger sink demand for reproduction in the OW cohorts (Fig. 4b, c, Supplementary Data 2).

We also performed time-series transcriptome analysis during the GS and OW sink-removal experiments at −1, 0, 1, 2, 4, 6, and 8 and −5, 0, 1, 2, 4, 6, 8, and 12 weeks relative to treatment initiation, respectively (Fig. 4b, c). Samples collected 1 week after leaf expansion corresponded to −1 and −5 weeks (prior to treatment initiation) for the GS and OW experiments, respectively. At 0 weeks, samples were collected 1 h after treatment initiation. Although we conducted transcriptome analysis for all time points unless leaves were obtained, DEG analysis was performed for the time points after treatment initiation when leaves from both treatments were available (0–8 and 0–4 weeks for the GS and OW experiments, respectively). We defined upregulated and downregulated genes in the senescence-enhanced (sink+) compared to senescence-delayed (sink−) treatments. In the OW sink-removal experiment, we detected 689 and 1141 genes overall (Fig. 4d, Supplementary Data 6) and 96 and 14 senescence-related genes (Fig. 4e) as upregulated and downregulated DEGs (in sink+ relative to sink−), respectively. We found a higher number of downregulated DEGs overall, while more senescence-related genes were upregulated (Fig. 4d, e). In the GS sink-removal experiment, we detected 60 and 63 genes in total, and two and zero senescence-related genes as upregulated and downregulated DEGs, respectively (Fig. 4f, Supplementary Data 6). The smaller number of DEGs in the GS cohort indicates that the responses of gene expression to sink-removal were minimal during summer. The number of enriched GOs (FDR < 0.05) was 0/2 (upregulated/downregulated DEGs) and 70/104 in the GS and OW sink-removal experiments, respectively (Supplementary Data 7).

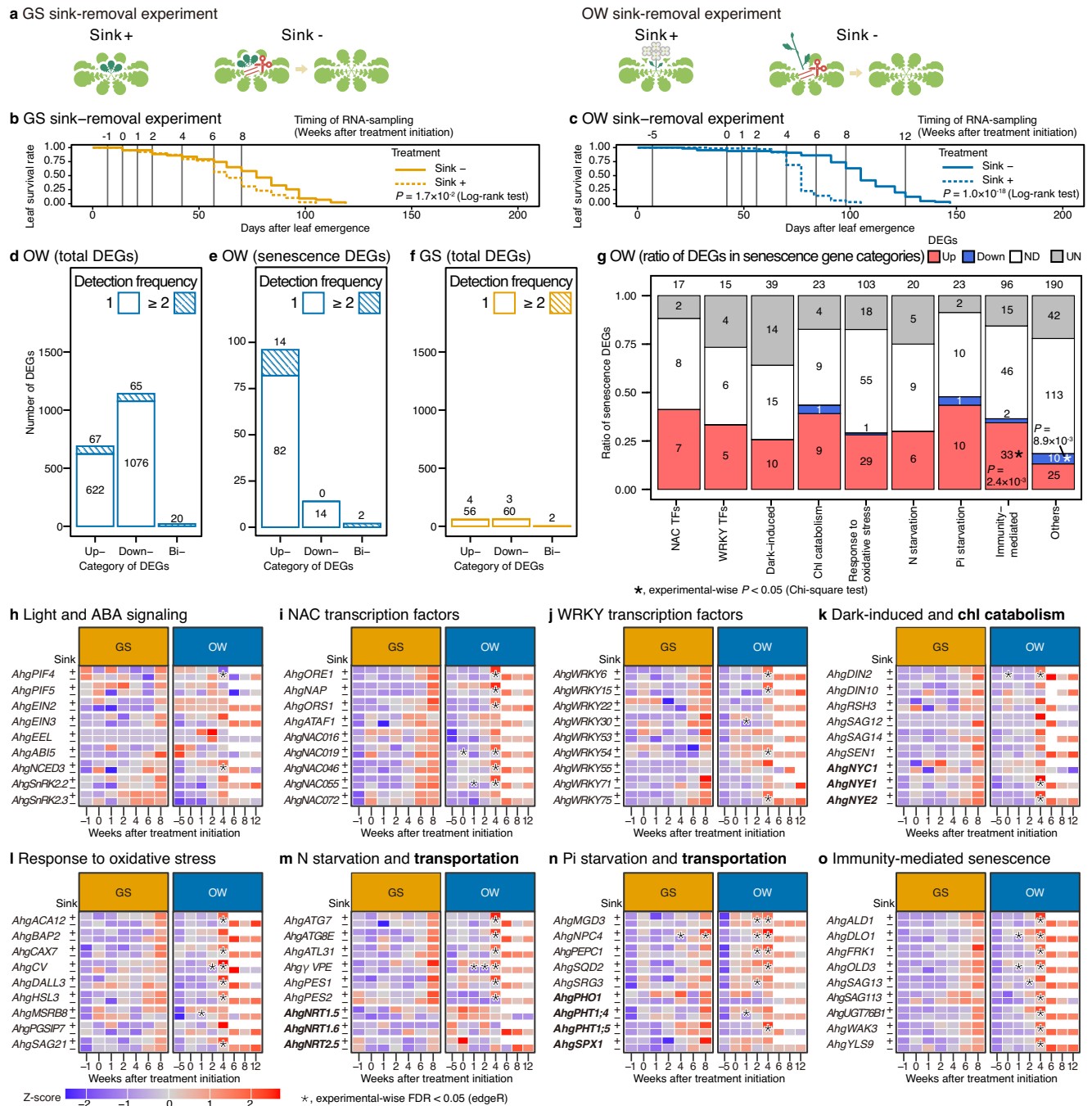

**Fig. 4 | Season-specific sensitivity of leaf senescence to sink demands in the GS and OW sink-removal experiments. a** Treatment of sink-removal experiments, with representative GS and OW cohorts. Sink refers to new leaves and flowering stalks, respectively. Responses of leaf longevity in the GS (**b**) and OW (**c**) sink-removal experiments. A number of upregulated, downregulated, and bidirectionally-regulated DEGs in the OW (total DEGs, **d**; senescence-related DEGs, **e**) and GS (total DEGs, **f**) self-shading experiments. We designated higher expression in the senescence-enhancing treatment (i.e., sink +) as upregulation. **g** Ratio of senescence-related DEGs (figures = numbers) for nine categories. Heat maps showing the time-series of gene expression under the sink+ (+) and sink- (−) treatments in the GS and OW cohorts. Genes listed are the same as **h**–**o** in Fig. 3. For other diagram details check the legend of Fig. 3. All statistics are two-sided. The multiplicity of the tests is corrected by the Bonferroni method (**g**–**o**). Source data are provided as a Source Data file.

Among the categories of senescence-related genes, nearly half of the Pi starvation senescence-related genes were upregulated in the OW sink-removal experiments (Fig. 4g). In addition, genes classified as NAC transcription factors, chlorophyll catabolism, nitrogen starvation, and immunity-mediated senescence-related genes showed relatively high proportions of upregulated DEGs (Fig. 4g). During the OW sink-removal experiment, almost no senescence-related genes were detected as significant DEGs until 2 weeks after treatment initiation,

when some Pi starvation and transportation genes were detected as upregulated DEGs (Fig. 4h–o). Sudden responses of upregulated senescence-related DEGs were detected at the very end of the leaf lifespan for the sink+ treatment (4 weeks after treatment initiation and 10 weeks after leaf emergence), and the leaves withered before the next sampling, indicating rapid translocation to the reproductive organs. The upregulated DEGs included many NACs (Fig. 4i), WRKYs (Fig. 4j), chlorophyll catabolism-related genes (Fig. 4k), oxidative

stress response genes (Fig. 4l), nitrogen and Pi starvation-induced transportation genes (Fig. 4m, n), and immunity-mediated senescence genes (Fig. 4o). Immunity-mediated senescence genes included homologs of *AGD2-LIKE DEFENSE RESPONSE PROTEIN 1 (AhgALD1)*, *DMR6-LIKE OXYGENASE 1 (AhgDLO1)*, *FLG22-INDUCED RECEPTOR-LIKE KINASE 1 (AhgFRK1)*, *ONSET OF LEAF DEATH 3 (AhgOLD3)*, *AhgSAG13*, *AhgSAG113*, *UDP-DEPENDENT GLYCOSYLTRANSFERASE 76B1 (AhgUGT76B1)*, *WALL ASSOCIATED KINASE 3 (AhgWAK3)*, and *YELLOW LEAF-SPECIFIC GENE 9 (AhgYLS9)* (Fig. 4o). The involvement of a homologue of Pi starvation-specific WRKY (*AhgWRKY6*)[31] suggests increased phosphorus remobilisation for seed production in the synchronised reproductive senescence of the OW cohort.

In the GS sink-removal experiment, although sink+ accelerated leaf senescence by 9.7 d in $L_{50}$, few senescence-related genes were detected as DEGs (Fig. 4h–o). This was partly due to the large variation in gene expression between replicates (PCA in Supplementary Fig. 5c), suggesting that uncontrolled determinants other than sink-removal contributed to variations in the transcriptome between replicates. Despite the larger variation between replicates in the GS experiment, we still observed considerable overlap between treatments during the GS experiment in the PCA (Supplementary Fig. 5c). In contrast, in the OW experiments, the transcriptomic status of sink+ changed quickly along the PC1 axis in the PCA space within 4 weeks, whereas sink-removal slowed the changes corresponding to strongly delayed senescence (Supplementary Fig. 5d). At the transcriptomic level, this suggests a smaller effect of sink demand on new leaf formation than on reproductive translocations.

Since sink-removal treatments cause wounding in experimental plants, the effect of sink-removal on gene expression could be partially due to a systemic response to wounding stress. Within genes with the 'response to wounding' GO (954 genes), the number of genes upregulated in sink (+) was larger than that of downregulated genes, which was contrary to the prediction that wounding response genes would show higher expression in sink (-) in response to sink-removal (Supplementary Fig. 6). In all experiments, including self-shading and sink-removal, more wounding response genes were consistently upregulated in the senescence-enhanced treatments (Supplementary Fig. 6). Thus, the positive effect of sink-removal on leaf longevity may not be derived from wounding stress, although we cannot exclude the possibility that a specific set of genes responds to wounding systemically.

## The difference in leaf senescence regulation between GS self-shading and OW sink-removal

Comparisons between all sets of experiments revealed the strong effects of GS self-shading and OW sink-removal as secondary controls of leaf senescence, in addition to seasonal effects. In the GS self-shading, OW self-shading, GS sink-removal, and OW sink-removal experiments, the upregulated DEGs (of 19,292 expressed genes) were 1015, 131, 60, and 689, respectively (Fig. 5a), whereas the downregulated DEGs were 1579, 77, 63, and 1141, respectively (Fig. 5b). When comparing GS self-shading and OW sink-removal, 277 and 645 DEGs were commonly upregulated and downregulated, respectively (green shading in Fig. 5a, b). However, a considerable number of DEGs were unique to each experiment, i.e., 684 and 395 upregulated and 926 and 477 downregulated DEGs were unique to the GS self-shading and OW sink-removal treatments, respectively (orange and blue shading, respectively, in Fig. 5a, b), suggesting that, in addition to the shared responses, leaves responded differently between the GS self-shading and OW sink-removal experiments.

GO enrichment analysis of these genes indicated the involvement of many abiotic stresses, defence responses, and some senescence-related GOs for the upregulated DEGs (Fig. 5c, Supplementary Data 8), and many photosynthesis-related and abiotic stress GOs for the downregulated DEGs (Fig. 5d, Supplementary Data 8). Specific involvement of senescence- and photosynthesis-related GOs only for

upregulated and downregulated DEGs, respectively (Fig. 5c, d), is likely to represent senescence activation and suppression of photosynthesis during senescence regulation. Particularly, 'leaf senescence' GO was strongly enriched in the commonly upregulated DEGs, but the enrichment of the same GO term was significant for the DEGs unique to the GS self-shading and OW sink-removal experiments (Fig. 5c), indicating the presence of a distinctive set of senescence-related DEGs unique to GS self-shading or OW sink-removal. Not all upregulated DEGs necessarily represent senescence controls, and the results indicate the necessity for further studies on the involvement of response mechanisms in abiotic stress and defence mechanisms in senescence controls triggered by distinctive cues.

In the comparison between GS self-shading and OW sink-removal for 432 senescence-related genes, we found equivalent numbers of shared and unique DEGs, especially for upregulated DEGs (Fig. 5e, f), suggesting that, in addition to the shared senescence controls, distinct senescence control mechanisms operate depending on the experiments. For instance, commonly upregulated DEGs were characterised by the inclusion of *NAC* transcription factors, chlorophyll catabolism, and nitrogen and Pi starvation-related genes (Fig. 5e). Unique sets of WRKYs were selected as upregulated DEGs: *AhgWRKY22* and *AhgWRKY71* were specific to the GS self-shading experiments, whereas *AhgWRKY6*, *AhgWRKY15*, *AhgWRKY30*, and *AhgWRKY54* were specific to the OW sink-removal experiments (Figs. 5e, 3j, 4j). Comparing uniquely upregulated DEGs, the GS self-shading experiment was characterised by the dominance of dark-induced senescence genes, whereas the OW sink-removal experiment was characterised by the dominance of immunity-mediated senescence genes and involvement of unique phosphorus starvation-responsive senescence genes (Fig. 5e). Many of the downregulated senescence-related DEGs were unique to the GS self-shading experiment (Fig. 5f).

## Comparisons of time-series transcriptomes between intact GS and OW leaves

As the sink+ treatment in the sink-removal experiments was the result of intact individuals, the time-series transcriptome data allowed us to compare GS and OW cohort leaves that showed short and extended longevities. We compared the transcriptomes of intact GS and OW leaves at 1, 6, 8, and 10 weeks after leaf expansion, although transcriptome differences should represent senescence and all other responses between the GS and OW cohort. In the total transcriptomes, 3758 genes were detected as DEGs at least once between the GS and OW cohorts when compared at the same age (1, 6, 8, and 10 weeks after leaf emergence; Supplementary Data 9). Among these, 1906 and 1669 genes were upregulated in GS and OW, respectively, either once or multiple times in a consistent direction. The former was enriched by leaf senescence, heat, drought, herbivory, and infection stress-related GOs, whereas the latter was enriched by cold response-, membrane-, and light response-related GOs (Supplementary Data 10). Most GOs, except for leaf senescence, probably reflected the seasonal temperature regime experienced by the GS and OW cohorts. Therefore, we compared only senescence-related genes between the GS and OW cohorts across the entire time-series (Fig. 6).

Expression patterns of senescence-related genes were most distinctive between emerged (1-week-old) and senesced (10-week-old) leaves and were similar between the GS and OW cohorts in the cluster analysis and PCA (Fig. 6a, b), suggesting that the initial and final states of leaf senescence are similar, irrespective of the season. The GS and OW cohorts were distinct in their expression patterns of senescence-related genes at 6 and 8 weeks after emergence, presumably representing the distinctive controls of leaf aging between the GS and OW cohorts. The cluster analysis showed that the transcriptome states of the OW cohort at 6 and 8 weeks remained closer to the initial states, whereas those of the GS cohort were closer to the final states, although PCA showed that they were not only the

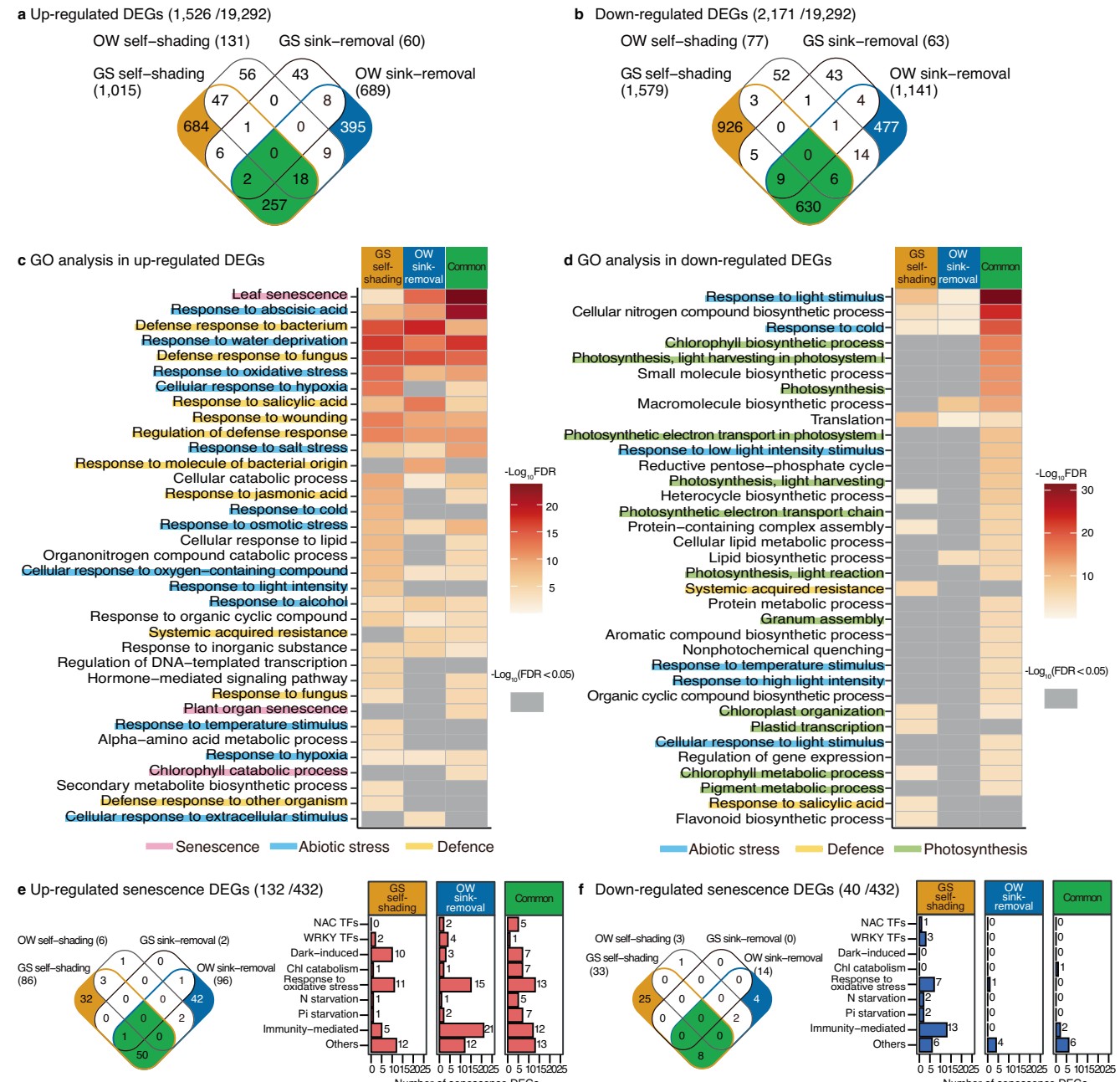

**Fig. 5 | Shared and unique DEGs in a series of manipulation experiments.** Venn diagrams of upregulated (**a**) and downregulated (**b**) DEGs across the manipulation experiments. We designated higher expression in the senescence-enhancing treatment (i.e., shaded and sink+) as upregulation. Most of the significantly upregulated and downregulated genes were detected once or multiple times in the time-series sampling with identical directions, and we treated these genes as upregulated and downregulated DEGs. The results of 1526 and 2171 upregulated and downregulated DEGs, respectively, in 19,292 expressed genes (average expression, log$_2$ (rpm + 1) > 1) are shown separately. In **a**, **b**, GS self-shading and OW sink-removal experiments are compared with their differences appearing as orange and blue regions, respectively, that represent DEGs unique to each experiment. Green region indicates DEGs shared by both experiments. GO analysis for upregulated (**c**) and downregulated (**d**) DEGs unique to GS self-shading, those unique to OW sink-removal, and those common to both experiments. Results of the top 35 enriched GOs are listed, and GO terms related to senescence, abiotic stress, defence, and photosynthesis are colour shaded. Venn diagrams of upregulated (**e**) and downregulated (**f**) senescence-related DEGs across the manipulation experiments. The bars show the number of DEGs in the nine senescence categories for the unique and shared ones in the two sets of experiments. Source data are provided as a Source Data file.

early and late states on the same trajectory of transcriptome changes (Fig. 6a, b).

We also conducted k-means clustering across senescence-related genes that recognised four clusters, depending on the differences in the time-series patterns of gene expression (indicated by coloured bars in Fig. 6a). Approximately half of the genes were in Cluster 1, which showed relatively similar time-series patterns between the cohorts and were strongly upregulated at 10 weeks. These genes were widely distributed across different categories of senescence-related genes (Fig. 6c). Clusters 2 and 3 showed different expression patterns in the GS and OW cohorts at 6–8 weeks (Fig. 6a). Cluster 2 was characterised by upregulation in the GS cohort and included three NAC transcription factors: *AhgATAF1*, *AhgNAC003*, and *AhgNAC032* (Fig. 6a, c). None of these NACs were detected as DEGs in any of the self-shading or sink-removal experiments. Therefore, they may be important for the seasonal acceleration of senescence in the GS

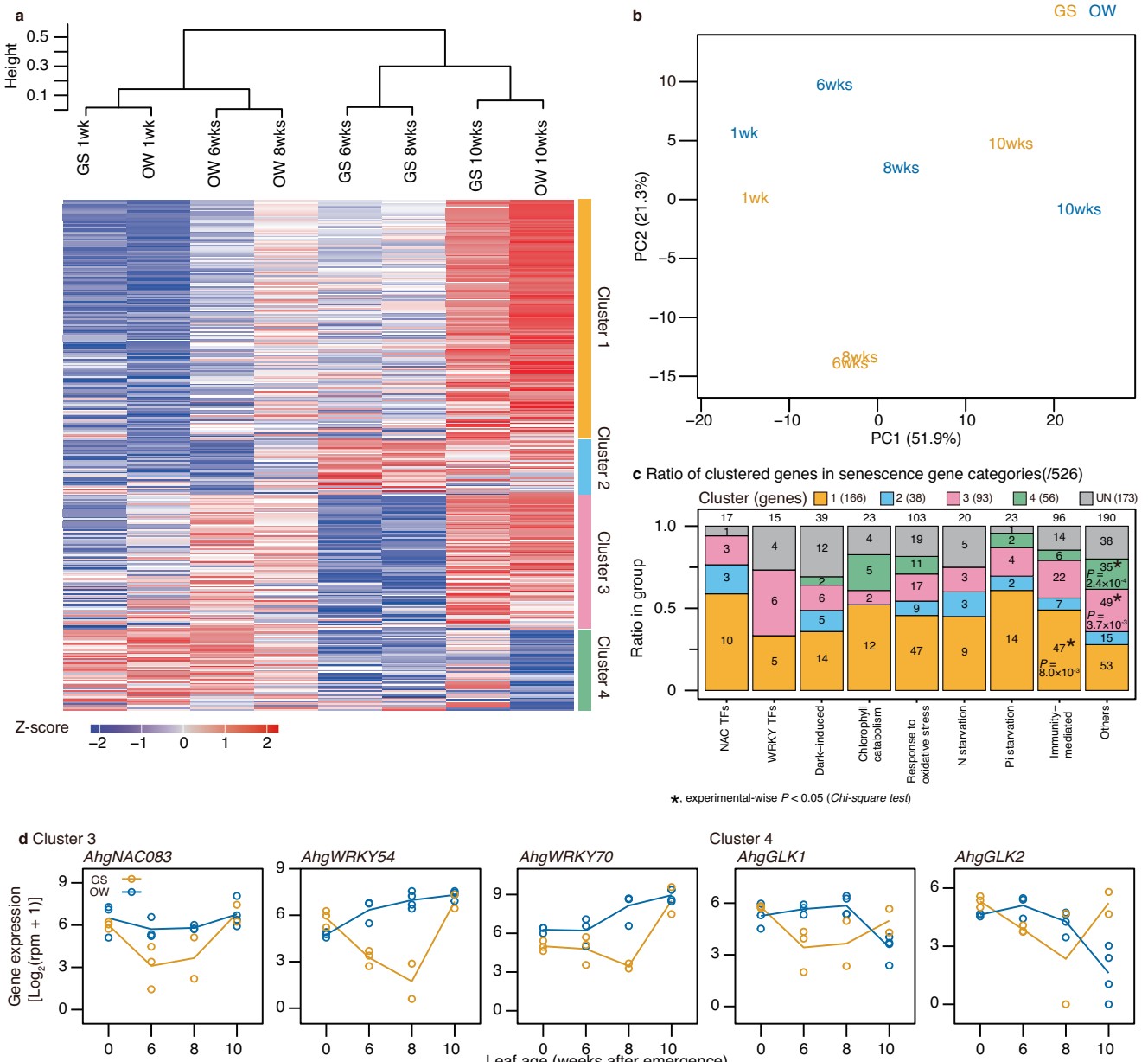

**Fig. 6 | Comparison of GS and OW intact leaves after 1, 6, 8, and 10 weeks of leaf emergence. a** Gene expression patterns of senescence-related genes in heat maps (shown for 353 expressed genes out of 432) and cohort-age combinations were arranged by the Ward method clustering (top of the diagram). Genes were arranged vertically based on the results of k-means clustering (left of the diagram).
**b** Principal component analysis based on the gene expression of the senescence-related genes (353 genes). **c** Ratio of senescence-related genes in different k-mean clusters (gene expression patterns) for nine categories of senescence-related genes. The number of genes are shown by figures. UN indicate 'unexpressed' genes. **d** Gene expression patterns of representative negative regulators of leaf senescence, that is, *AhgNAC083*, *AhgWRKY54*, and *AhgWRKY70* identified in Cluster 3, and *AhgGLK1* and *AhgGLK2* identified in Cluster 4, which showed upregulation in the OW cohorts during senescence. These analyses were performed using the transcriptome data of the sink+ treatments in the GS and OW sink-removal experiments. All statistics are two-sided. The multiplicity of the tests is corrected by the Bonferroni method (**c**). Source data are provided as a Source Data file.

cohorts. *ATAF1* and *NAC032* in *A. thaliana* are oxidative stress-induced positive regulators of leaf senescence[32,33]. Cluster 3 was characterised by upregulation in the OW cohort and included three NACs and six WRKYs, i.e., *AhgNAP*, *AhgNAC017*, *AhgNAC083*, *AhgWRKY1*, *AhgWRKY28*, *AhgWRKY42*, *AhgWRKY53*, *AhgWRKY54*, and *AhgWRKY70* (Fig. 6a, c, d). *AhgNAC083*, *AhgWRKY54*, and *AhgWRKY70* are known negative regulators of leaf senescence[34,35]. Cluster 4 was characterised by upregulation at 1 week both for GS and OW, but its upregulation was only maintained until 8 weeks for the OW cohort and included homologues of *GOLDEN2-LIKE1* (*AhgGLK1*) and *AhgGLK2* (Fig. 6d). These are all known as negative regulators of senescence that maintain chloroplasts by antagonising *ORE1*[36]. The upregulation of negative

regulators of senescence in OW plants was consistent with increased leaf longevity during winter.

## Discussion

Our study showed that the evergreen strategy of perennial *Arabidopsis* exhibits a distinct switching of foliage functions between the growing and overwintering seasons, which is mediated by the control of leaf senescence (Fig. 7). It turned out that leaf senescence is phenotypically and transcriptomically shut down during winter. Our experiments suggest that age-dependent and shade-induced senescence halts during winter. Regarding the control of leaf senescence, our study allowed us to define two types of leaf cohorts (GS and OW) and three

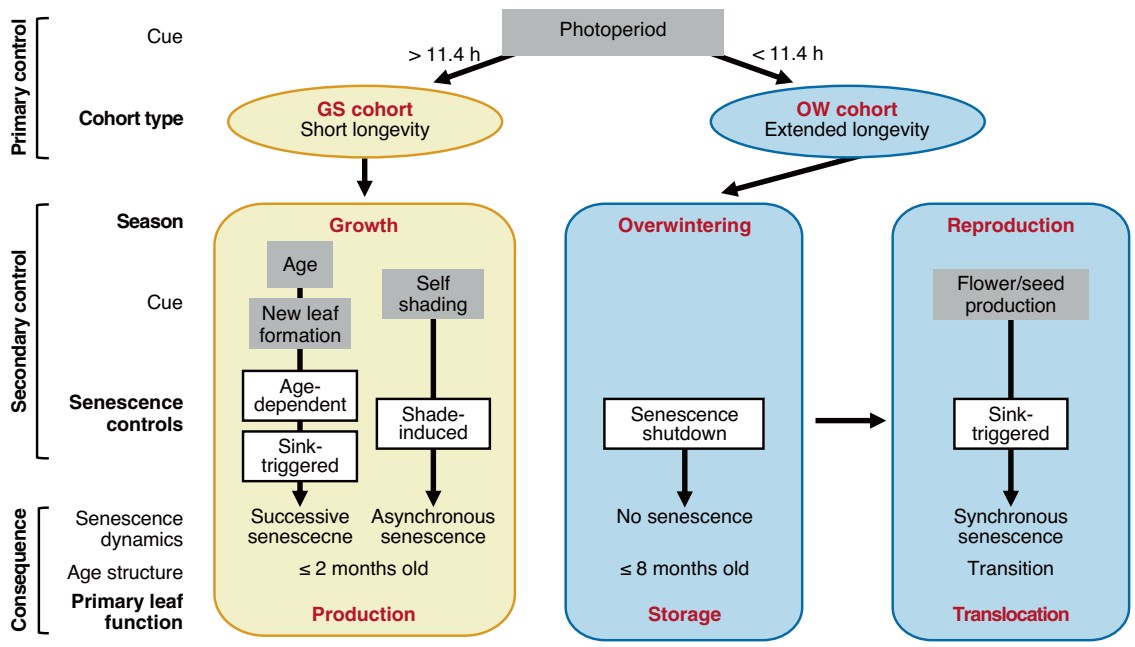

**Fig. 7 | Schematic diagram representing seasonal controls of the evergreen *Arabidopsis* estimated in this study.** Two types of leaf cohorts (GS and OW) and three distinct periods (growth, overwintering, and reproductive) with different functions (production, storage, and translocation, respectively) are realised by the two-layered control of leaf senescence.

distinct periods, i.e., growth, overwintering, and reproductive seasons (Fig. 7).

OW cohorts were well-separated from GS cohorts by photoperiod rather than temperature, indicating that photoperiod provides the primary control of leaf senescence in the evergreen lifecycle (Fig. 7). In *A. thaliana*, leaf senescence is accelerated under long-day conditions compared to short-day conditions, and *ORE1* is upregulated under long-day conditions through the regulation of a circadian clock component, GIGANTEA (GI)[37]. Another point for the importance of photoperiod in controlling leaf senescence comes from annual plants[38]. It has been suggested that onset of leaf senescence is accelerated by long day-lengths in long-day plants, such as *A. thaliana*[39] and *Hordeum vulugale*[40], and by short day-lengths in short-day plants, such as *Glycine max*[41].

The halting of leaf senescence in an evergreen plant during winter is a finding in this study, and transcriptomic analysis suggests that the extended leaf longevity during winter is not just a passive effect of low temperature, but rather an active halting of age- and shade-induced senescence. We observed the upregulation of negative regulators of leaf senescence, such as *AhgNAC083*, *AhgWRKY54*, *AhgWRKY70*, *AhgGLK1*, and *AhgGLK2*, during winter in the OW cohort (Fig. 6d). Further experimental studies are needed to elucidate the mechanisms underlying the shutdown of leaf senescence during winter, particularly in relation to photoperiod. The developed age structure of the overwintering rosettes is likely to serve primarily for storage, and low leaf turnover is expected to minimise storage loss (Fig. 1h, i). The cessation of shade-induced senescence results in a rosette structure in which the lower leaves are covered by the upper leaves during winter. Storage in covered leaves may be safeguarded by exposing the upper leaves to frost damage caused by ice-cold wind and radiation chilling, and we observed such protective effects in the upper leaves under field conditions. In a previous study using an evergreen tea, *Camellia sinensis*, it was reported that leaves in growth and dormant season showed altered gene expression in cytokinin-, ethylene-, auxin-, and gibberellin-mediated pathways, which may ultimately prevent abscission of winter leaves[42]. Further studies are required to determine the extent to which winter abortion of leaf senescence and its underlying mechanisms are common among different plant species.

During the growing season, activated age-dependent senescence is considered to realise high leaf turnover, presumably maximising photosynthetic production at the whole-plant level (Fig. 7). Previous studies on *A. thaliana* and other crops have defined this type of senescence as successive senescence during vegetative growth. We found that successive senescence occurs in terms of cohort averages, but that leaf longevity within cohorts becomes quite variable because of the shade-induced control of senescence in response to the local light environment of individual leaves. This resulted in a rather peculiar pattern of leaf longevity, i.e., short but variable longevity of the GS cohort. The selective thinning of shaded leaves is expected to further optimise the rosette structure for enhanced photosynthetic assimilation. The active state of shade-induced senescence was clearly detected in the transcriptomic responses. Nine genes upregulated in the later stage of the shade treatment in our study are on the list of late upregulated genes after dark-induced senescence becomes irreversible in *A. thaliana*[43]. We also detected upregulated DEGs related to chlorophyll catabolism and Pi and nitrogen starvation in response to self-shading, and metabolic responses have been reported in individually darkened leaves of *A. thaliana*[44]. Upregulation of the *ORE1* homologue and downstream senescence-activated genes is known to function in age-dependent and/or shade-induced pathways. Although some upstream transcription factors showed weak responses, the strong response of *AhgORE1* may be the result of multiple feed-forward loops involving *ORE1*, identified in *A. thaliana ORE1*[11].

The removal of the reproductive sink largely delayed leaf senescence, suggesting that whole-plant synchronisation of leaf senescence during reproduction is due to the strong nutrient demand for flower, fruit, and seed production (Fig. 7). A strong sink demand for reproduction was also detected at the transcriptome level, and genes related to nitrogen and Pi starvation and transportation were upregulated during reproduction (Fig. 4m, n). Seeds typically increase their phosphorus concentration synchronously with a decrease in the phosphorus of senescing leaves[45]. Removal of reproductive structures or prevention of reproduction was speculated to have limited effect on

leaf longevity in *A. thaliana*[19], however another study in *A. thaliana* showed that the activity of ascorbate peroxidases, major ROS scavengers, was transiently reduced during bolting, and that removal of flowering stalks at an early stage resulted in an increase in leaf longevity[46]. In *A. halleri*, we observed a significant effect on leaf longevity from the reproductive sink. Since a perennial life history requires resources for vegetative growth after reproduction, controlling leaf senescence in response to reproductive demands may optimise resource allocation between reproduction in the current and future vegetative growth. Reproductive senescence is unique due to its involvement in immune-triggered senescence. Comprehensive transcriptomic analysis of leaf senescence in *A. thaliana* revealed the involvement of many defence genes in leaf senescence and bolting, and an increase in salicylic acid levels at the later stages of leaf senescence[20,24]. As the immune response is characterised by systemic signalling, it may enhance the synchronised senescence of leaves of different ages in a single plant.

In conclusion, the two-layered regulation of leaf longevity at the level of individual leaves achieves the switching of foliage structures with different functions at the whole-plant level, i.e., photosynthetic production during the growth season, storage during winter, and the translocation of resources for reproduction in spring (Fig. 7). Seasonal switching of leaf senescence control in evergreen plants may represent a fundamental strategy for plants growing in temperate and boreal regions, which is as important as deciduousness.

## Methods

### Study species
The study species *A. halleri* (L.) O'Kane & Al-Shehbaz subsp. *gemmifera* (Matsum.) O'Kane & Al-Shehbaz is distributed in East Asia and the Russian Far East[47,48]. Plants form rosettes throughout the year, except during the flowering season (March–May in central Japan), when the flowering stalks are elongated and leaves on the stems are raised as a result of stem internode elongation[25]. The flowers are self-incompatible and pollinated by small bees, flower flies, and small butterflies, and the seeds mature and disperse in June[25]. After the flowering season, shoot apical and lateral meristems form aerial rosettes on the flowering stems (reversion)[25]. The formation of aerial rosettes reflects vegetative reproduction[25].

### Study site
The study site was located downstream of an abandoned mining site in Hyogo Prefecture, central Japan (35°10′N, 134°93′E, altitude ca. 200 m, Omoide-gawa study site)[25]. A field study was conducted with the consent of the local community. The studied species is a metallophyte often found in soils contaminated with heavy metals[48,49]. At this site, *A. halleri* grows along a stream running through an open secondary forest, but the vegetation density here was low because of heavy metal contamination. No other Brassicaceae species were detected at the site. A leaf phenology study was conducted in a 20 × 25 m rectangular plot (main plot) established in 2005 to study the long-term phenology of *A. halleri*[25]. The plot was subdivided by grid lines at 1-m intervals, and all intersections of the grid lines (grid points) were marked with bricks[25].

### Selection of monitoring plants
During the 4-year field study, 30 monitoring plants were selected every year in the plot by establishing twelve 3 × 3 m subplots within the main plot where *A. halleri* was relatively abundant. However, the same set of individuals was used in the 2019–2020 survey. One to three plants were selected within each subplot at a distance of >1 m between plants. The selected plants had more than four leaves (an average of 18 leaves) and no signs of obvious damage. The dates of initiation of bolting, flowering, and reversion were recorded. Events were defined as the elongation of the main stem >5 mm, opening of the first flower, and

formation of the first aerial rosette. The phenological measurements of individual leaves were carried out. Lost plants were replaced with nearby plants ($n = 6$, 6, and 3 during the first-, second-, and third-year censuses, respectively). The number of plants measured per census ranged from 28 to 30, except during the snow cover on 30 January 2018 ($n = 23$) and flooding on 14 July 2020 ($n = 23$). No individuals were replaced during the fourth year, with 15 individuals surviving until the end of the census.

### Leaf phenology
The emergence, growth, and longevity of individual leaves were determined for biweekly leaf cohorts that emerged from 10 October 2017 to 21 September 2021 by tagging individual leaves every 2 weeks, except twice when snow cover prevented access to the plants. The fate of these leaves was recorded weekly until the leaves of the last cohort withered on 11 January 2022. In each biweekly census, all the new leaves of each individual that emerged after the last census were counted and recorded. For each plant, we tagged the two smallest young leaves at that stage that were still folded and <5 mm in length. If there were three or more leaves, the two tagged leaves were treated as representatives of the other leaves in age structure analysis (details described later). If no new leaves were present, no leaf was tagged. A group of leaves that emerged within 2 weeks of the previous census to a census date was defined as a leaf cohort (leaf populations of the same age and emergence date was set to the census date). A total of 102 cohorts were examined during the entire census period (32.7 on average, range between 8–51 leaves/cohort; Supplementary Data 1). For each weekly census, we recorded whether each tagged leaf was alive or withered. Leaves were considered withered if they were either lost or completely senesced (i.e., when there was no green area on the leaf). The length (mm) of all tagged leaves was recorded every 2 weeks. Leaf length was defined as the distance between the most basal edge of the lamina and tip of the leaf, measured along the midrib. Leaf length measurements started from the 13 February 2018 cohort; there were no leaf growth data for the earlier cohorts.

### Survival and longevity analyses
Survival analysis was conducted by assuming a Weibull distribution for the longevity of individual leaves within a cohort. For each cohort, we fitted a survival curve (survival rate = exp $[-(x/\beta)^\alpha]$) to a Kaplan–Meier survival plot, where $x$ is the date after leaf emergence, to estimate $\alpha$ and $\beta$. $L_{50}$ and $\alpha$ were calculated to estimate the leaf longevity of a cohort and degree of synchronisation within a cohort (Supplementary Data 1)[27].

### Leaf growth period and rate
The maximum leaf length was defined as the maximum value among repeated measurements of the length of individual leaves from emergence to death. Mature leaf length was defined as the leaf length at which it first exceeded 90% of the maximum leaf length. The leaf growth period was defined as the number of days required to reach the mature leaf length. To calculate the leaf growth rate, the mature leaf length was divided by the leaf growth period. Leaves with a maximum leaf length <10 mm were excluded from the calculations of the leaf growth period and rate. The cohort means of the parameters described above are listed for all cohorts in Supplementary Data 1.

### Age structure analysis of extant and withered leaves
Cohort composition was analysed for the total leaf population of all plants examined ($n = 30$). The number of extant and withered leaves in the foliage population was estimated for each cohort by multiplying the ratio of extant to withered tagged leaves by the total initial number of leaves for the corresponding cohort. Age structures were determined every 2 weeks for extant and withered leaves in the foliage population. For census dates when the number of plants examined was

<30, the values were adjusted by replacing missing individuals with the average leaf number and age structure calculated for the extant plants.

## k-means clustering of cohorts and DTA of cohorts

Based on $L_{50}$ and leaf growth rate, k-means clustering was performed to divide leaves into groups using the R version 4.0.2[50] with the package "cluster" (v2.1.4) [51]. The optimal number of clusters was estimated using the silhouette method[25]. Environmental factors contributing to the separation of the GS and OW leaf cohorts were analysed by DTA using the C5.0 decision tree machine-learning algorithm[52,53] with the package "C5.0" (v0.1.4)[54] of R. C5.0 identifies the factors that divided the data into smaller portions to maximise the purity of the terminal nodes based on entropy. We constructed a tree model to predict the classes of the GS and OW cohorts based on environmental factors. Cluster analysis was not applied to cohorts before January 2018 because of a lack of growth rate data.

## Self-shading experiments

The GS and OW self-shading experiments were conducted on 2 July 2019 and 5 January 2021, respectively. For each experiment, 30 individual plants separated by >1 m were selected from the natural population. For each rosette, the six youngest leaves that were <10 mm in length were tagged with threads, and we applied shade and exposure treatments 3 weeks after tagging. Six tagged leaves were divided into two equal groups of three for each treatment. For the shade treatment, a neighbouring leaf was moved and fixed with thread to completely cover one of the tagged target leaves. In exposure treatment, the target leaves were completely exposed by adjusting the position of the neighbouring leaves using threads. No target or neighbouring leaves were damaged by these manipulations. During every weekly visit, we readjusted the positions of neighbouring leaves to maintain the treatments. Both the shaded and exposed treatments were set on the same individuals in the self-shading experiments (Supplementary Fig. 7).

## Sink-removal experiment

The GS and OW sink-removal experiments were conducted on 28 July 2020 and 8 February 2022. We selected 48 and 64 plants, comprising 24 and 32 pairs of similar-sized neighbouring plants <1 m from each other, respectively. One of the paired plants was assigned to the sink-removal treatment (sink-) and the other to the intact control (sink+). Three young leaves per plant were tagged using thread. For the GS and OW sink-removal experiments, new leaves and flowering stalks were respectively excised at the base using scissors. New leaf removal in the GS sink-removal experiment started 1 week after tagging. In the OW sink-removal experiment, flowering stalk removal began on 22 March 2022, when all the selected plants had started bolting. Sink-removal was repeated weekly until the end of the experiment. During the first 3 weeks, flowering stalks were removed every 3–4 d to prevent increased translocation due to rapid regrowth. In the control group, plants were touched and handled at a similar intensity as in the sink-removal treatments. Three leaves per individual were prepared for each treatment, and sink+ and sink-treatments were set in different individuals in the sink-removal experiments (Supplementary Fig. 8).

## Measurement of leaf longevity in the experiments

In all experiments, one of the three replicated leaves per individual for each treatment was designated as a candidate leaf for RNA sampling (Supplementary Figs. 7 and 8). The fate of all tagged leaves was recorded weekly and used for leaf longevity analysis, except for those sampled for RNA-seq (described later in 'RNA sampling'). For plants that no leaves were removed from for RNA sampling, the records of all leaves were used for leaf longevity analysis. We used records of the remaining leaves for leaf longevity analysis of the plants from which one leaf was taken for RNA sampling during the experiments (Supplementary Figs. 7 and 8). Leaf longevity records between plants with and without RNA sampling did not differ significantly (data not shown), and the sampling of two and one leaves per plant once during the self-shading and sink-removal experiments, respectively, had minor effects on leaf longevity analysis. Although the removal of new leaves affected the longevity of the remaining leaves in the sink-removal experiment, the removal of new leaves for the GS sink-removal treatment was conducted weekly, and RNA sampling was performed once per plant during the experiments. We also excluded leaves that withered before treatment initiation and those removed by deer herbivory (this procedure lengthened the $L_{50}$ estimates compared to those estimated in the leaf phenology analysis).

Consequently, in the GS self-shading experiment, we used the records of 60 and 58 leaves from 29 plants in the shaded and exposed treatments, respectively (Supplementary Fig. 7). In the OW self-shading experiment, we used records of 60 and 63 leaves from 30 and 29 plants for the shaded and exposed treatments, respectively (Supplementary Fig. 7). In the GS sink-removal experiment, we used the records of 39 and 43 leaves from 18 and 22 plants in the sink+ and sink-treatments, respectively (Supplementary Fig. 8). In the OW sink-removal experiment, we used records of 71 and 65 leaves from 30 and 32 plants each for the sink+ and sink- treatments, respectively (Supplementary Fig. 8). The differences in the time-series of leaf longevity between the experimental treatments were compared using the log-rank test (time-stratified Cochran–Mantel–Haenszel test) with the package "survival" of R[50]. $L_{50}$, $\alpha$ and $\beta$ were calculated for each treatment for all the experiments (Supplementary Data 2).

## RNA sampling

In the self-shading and sink-removal experiments, a time-series sampling of leaves was performed for RNA-seq analysis (Supplementary Figs. 7 and 8). One of three tagged leaves per plant was sampled once in the time-series. In the GS self-shading experiment, six and five treatment pairs were sampled at each of the following sampling times: 0, 2, 4, 6, and 8 weeks after treatment initiation (23 July, 6 August, 20 August, 3 September, and 17 September 2020, respectively). Four pairs of samples per time point were used for the RNA-seq analysis. In the OW self-shading experiment, five treatment pairs were sampled at each of four sampling times, i.e., 0, 4, 8, and 12 weeks after treatment initiation (26 January, 23 February, 23 March, and 20 April 2022, respectively). Five pairs of samples per time point were analysed for RNA-seq.

In the GS sink-removal experiments, samples were collected at seven-time points: 1 week before treatment, and 0, 1, 2, 4, 6, and 8 weeks after treatment initiation (28 July, 4, 11, and 18 August, and 1, 15, and 29 September 2020, respectively). Either two (at 6 and 8 weeks after treatment initiation) or three (at the other five time points) treatment pairs were sampled at each time point. All obtained samples were used for RNA-seq. In the OW sink-removal experiment, samples were collected at eight time points: 5 weeks before treatment, and 0, 1, 2, 4, 6, 8, and 12 weeks after treatment initiation (15 February, 22 and 29 March, 5 and 19 April, 2 and 17 May, and 14 June 2020, respectively). Either three (6 weeks after the start of treatment) or four (other time points) replicates per treatment were sampled at each time point. Pairwise samples were used up to 4 weeks after treatment, and only the sink- treatment was sampled at later time points. In the sink+ treatment, all leaves had withered within 6 weeks after treatment initiation, and therefore, no leaves were sampled after this point. All collected samples were subjected to RNA-seq analysis.

All samples were collected at noon and the sampled leaves were immediately placed in 1.5 mL microtubes containing 500 μL RNAlater (Thermo Fisher Scientific, Waltham, MA, USA, #AM7021r) on ice and transported to the laboratory. After overnight storage in a refrigerator

at 4 °C, they were stored at −20 °C in a freezer until they were used in further analyses.

## RNA extraction and library preparation

Leaf samples were homogenised in lysis/binding buffer using a multi-bead shocker (Yasui Kikai, Osaka, Japan). The mRNA was isolated directly from the homogenate using streptavidin magnetic beads (New England Biolabs, Ipswich, MA, USA, #S1420S) and 5′ biotinylated polyT oligonucleotide[55]. RNA libraries were prepared using the Breath Adapter Directional sequencing (BrAD-seq) method for strand-specific 3′ digital gene expression quantification[56]. Briefly, the mRNA was heat-fragmented and primed with a 3′ adapter-containing oligonucleotide primer targeting the polyA tail of the mRNA. cDNA was synthesised using RevertAid Reverse Transcriptase (Thermo Fisher Scientific, #EP0441) on a Veriti Dx Thermal Cycler (Thermo Fisher Scientific). The 5′ adapter was added by strand-specific breath capture and the second strand was synthesised using DNA Polymerase I (Thermo Fisher Scientific, #EP0041). Final PCR enrichment was performed using oligonucleotides containing the full adapter sequence, with a unique index for each sample.

PCR products were purified and selected using AMpure XP beads (Beckman Coulter, Brea, CA, USA, #A63881). The size distribution and concentration of the library were measured using a Model 2100 Bioanalyser (Agilent Technology, Palo Alto, CA, USA) and QuantiFluor DNA System (Promega, Madison, WI, USA) with an Infinite 200 PRO microplate photometer (TECAN, Basel, Switzerland), respectivly. Products from 40, 40, 38, and 48 samples from the GS self-shading, OW self-shading, GS sink-removal, and OW sink-removal treatments, respectively, were pooled as two sets of libraries for Illumina sequencing. The four libraries were sequenced in two lanes on a HiSeq 2500 instrument (Illumina, San Diego, CA, USA). The original BrAD-seq protocol[56] was modified to use KAPA HiFi HotStart ReadyMix (Kapa Biosystems, Woburn, MA, USA, #KK2062) for the final PCR.

## RNA-seq data analysis

The 50-base single-end reads with index sequences were determined using a HiSeq 2500 (Illumina) on the TruSeq v3 platform. The sequence data were deposited in a short-read database. Pre-processing and quality filtering were performed using trimmomatic (v0.36)[57]. The reference sequences used were the nuclear and chloroplast transcript sequences of *A. halleri*[58,59], 8,109 viral sequences NCBI (GenBank), and the ERCC (External RNA Controls Consortium) spike-in control (Thermo Fisher Scientific). Transcripts of *A. halleri* (32,553 genes)[58] were annotated using the BLAST best hit against Araport11[60]. The pre-processed RNA-seq reads were mapped to reference sequences and quantified using RSEM (v1.2.31)[61] and Bowtie2 (v2.3.4.1)[62]. The estimated read count for each gene was converted to $\log_2 (rpm + 1)$.

Genes with average $\log_2 (rpm + 1) \geq 1$ across samples within a set of experiments were used for subsequent analyses, which included 18,161, 18,174, 17,585, and 17,897 genes for the GS and OW self-shading and GS and OW sink-removal experiments, respectively. The DEGs between the treatments (shaded/exposed or sink+/sink-) were detected (experimental-wise FDR = 0.05)[63] using the package "edgeR"[64] in R[50] (Supplementary Data 4 and 6 for all genes). We also conducted DEG analysis between GS and OW sink+ plants at the same leaf age (1, 6, 8, and 10 weeks after leaf emergence, Supplementary Data 8). Paired sample models were used and multiple testing corrections were applied to set the FDR criteria.

## PCA on the gene expression data

PCA was done by the prcomp function in the R[50] for expressed genes (average $\log_2 (rpm + 1) \geq 1$ across samples). We used samples from all experiments in a single PCA and plotted the results of the four experiments separately (Supplementary Fig. 5). Analysis was performed using all the expressed genes (19,292 genes) (Supplementary Fig. 5).

In the analysis of the intact GS and OW cohorts, expressed senescence-related genes (353 genes) were analysed, and gene expression was averaged across replicates for each cohort and time point combination prior to PCA and clustering analysis (Fig. 6). Comparisons between time point and cohort combinations were conducted by hierarchical clustering using the correlation coefficient obtained by the cor function (method = 'spearman'), followed by hierarchical clustering using the hist function (method = 'ward.D2') of R[50]. The 353 expressed senescence-related genes were clustered using k-means clustering. The expression levels were z-scores transformed using the genescale function in the package "genefilter" (v1.82.1)[65] of R. The number of clusters was then determined by the clusGap function ($k = 4$), and clustering was performed with the kmeans function in the package "cluster"[51] of R[50].

## GO analysis

We conducted the GO analysis of the upregulated and downregulated genes separately in each experiment. GO terms were determined for the *A. halleri* genes that were successfully annotated as *A. thaliana* genes (30,162 of 32,553 genes). A correspondence table for the GO analysis was created using the ATH_GO_GOSLIM.txt.gz file (version 2023-03-31) from TAIR (https://www.arabidopsis.org/). Enrichment analysis was performed using Fisher's exact test with the fisher.test function of R[50]. *P*-values were corrected for multiple comparisons using the Benjamini–Hochberg method[63] with the p.adjust function of R[50].

## Statistics & Reproducibility

Statistical methods used in the study are described in detail in each corresponding section of Methods. For field observations, 4-year data were used to confirm reproducibility of seasonal patterns. No statistical method was used to predetermine sample size. The investigators were not blinded during field observations, experiments, and outcome assessment.

## Reporting summary

Further information on research design is available in the Nature Portfolio Reporting Summary linked to this article.

## Data availability

The RNA-seq reads data generated in this study have been deposited in the DNA Data Bank of Japan (DDBJ) database under the accession codes DRA013140 and DRA016957. The processed RNA-seq data are Supplementary Data 4–10. We used AGI code to refer *Arabidopsis thaliana* at TAIR (https://www.arabidopsis.org) and gene ID to refer *Arabidopsis helleri* subsp. *gemmifera* at Dryad (https://doi.org/10.5061/dryad.gn4hh). Source data are provided with this paper.

## Code availability

R codes used in this study were deposited in http://sohi.ecology.kyoto-u.ac.jp/AhgRNAseq/Data.zip.

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

## Acknowledgements

We thank Maasa Yumoto-Hamamura for support with fieldwork and chart design. This work was financially supported by JSPS Grant-in-Aid for Specially Promoted Research JP21H04977 (HK), JST CREST JPMJCR15O1 (HK), JSPS Grant-in-Aid for JSPS, Fellows JP19J22000 (GY), and JSPS Grant-in-Aid for Transformative Research Areas JP21H05659 (HN).

## Author contributions

H.K. and G.Y. designed this study. G.Y., J.S., and H.K. obtained data; G.Y., T.M., H.N., and M.N.H. analysed data; and G.Y. and H.K. wrote the manuscript with input from all authors.

## Competing interests

The authors declare no competing interests.
