## [Peer Review File · Nature Communications]

Seasonal switching of integrated leaf senescence controls in an evergreen perennial ArabidopsisREVIEWER COMMENTS

Reviewer #1 and #2 (Remarks to the Author):

By studying evergreen perennial plant *Arabidopsis halleri*, the authors identified factors that control leaf senescence in different seasons. Although the results are not that surprising, they are interesting, they fill a gap in the literature and improves our understanding on how senescence is coordinated. Although it is well known that regulation of senescence is very complex, this study illustrates well how complicated it is to make generalizations, as the developmental stage of the plant determines how it responds to environmental cues.

The study was conducted in the field over four years and the authors measured exhaustively the growth and longevity of leaves and identified two distinct seasons: growth season and overwintering season, that were separated by photoperiod. Shading and sink-removal treatments had different effects on leaf longevity in different cohorts indicating that the secondary controls for senescence differed between seasons. Shading delayed leaf senescence in the growth season, but not in the overwintering season. Leaf senescence at the end of winter season was on the other hand synchronized with flowering and the removal of sinks delayed leaf senescence more in overwintering season than in growth season. Overall, the leaf growth and longevity data have been thoroughly analysed, well presented and the results appear sound. In addition, the authors studied gene expression in different treatments in the two leaf cohorts and noted that senescence-associated gene (SAG) expression was off in overwintering leaves. I see a lot of potential in the generated time-series gene expression data; however, its presentation could be improved.

Methods are overall well described; however, I would recommend authors to better explain how the plants were sampled in the experiments. I do not really understand if the same plants sampled repeatedly during the time course over the season? And were the sampled plants included or excluded from the monitoring group for leaf longevity in the treatment plots? Since sink removal affected leaf longevity, especially in OW group, could leaf sampling affect leaf longevity as well? Were the shaded and exposed leaves sampled from the same plant individual?

Authors explained that in the sink-removal experiment, the control group was similarly disturbed by touching the plants. The effect of sink-removal could be partially due to the stress response induced by physical damage. Was there a control group for physical damage?

Transcriptomics results are focused on the well-established set of senescence-associated genes (SAGs) and the expression patterns of some of them are visualised in the heatmaps in figures 3 and 4. Although this is acceptable; it overlooks the other affected processes beyond the well-established change in the SAG expression. Differential expression analysis showed that thousands of genes were affected by the treatments in each leaf cohort. Authors compared treatment and control groups separately in each time point and cohort. However, it remains unclear to me how the DE genes were defined as up- or down-regulated in a time-series.

I would recommend authors to compare the complete lists of DE genes, not just the SAGs, between the cohorts and different treatments. The list of DE genes affected in each condition and if they are shared between any of the treatments could be added as supplementary information. It would be interesting to see what are the shared 266 up- and 636 down-regulated genes in GS self-shading and OW sink-removal treatments or the shared 64 genes up-regulated in self-shading treatment in both cohorts as indicated in supplementary figure 5? These analyses could be easily combined with enrichment analyses for the identified gene sets that could provide useful information of what kind of biological processes are involved in senescence regulation in each treatment and season.

What calls my attention is the high fold changes which are NOT significant especially in GS sink-removal group; this indicates high variation within the sample group. I would suggest that the authors to check the quality of their data before DE analyses and identify potential outliers. For example, clustering analysis or principal component analysis could be used to identify if there are outliers in the data.

Another thing is that there are several DE genes in all the treatments at the time point 0. Were the samples collected before the treatments started or after during the same day? Was this initial difference at time 0 considered in the subsequent analyses of the gene

expression data?

The treatments in GS and OW were performed two years apart, not during consecutive seasons. It would still be interesting to study what kind of processes are differently regulated between GS leaves and OW leaves in green leaves, during senescence process and in response to different treatments. This would provide some additional information on how OW leaves differ from GS leaves. Therefore, I believe that the study would benefit from additional analyses of the transcriptomics data that could perhaps shed more light into how senescence is induced or prevented during different seasons and in different treatments.

I would also recommend authors to extend the discussion of their results in line with existing literature. For example, in an evergreen perennial tea tree (*Camellia sinensis*), SAGs were not up-regulated in the winter leaves, instead the genes related to stress tolerance were highly expressed (Paul et al. 2014, DOI: 10.1038/srep0593). In addition, gene expression changes during dark-induced and developmental leaf senescence have been extensively studied with *Arabidopsis thaliana* and it has been shown that SAG expression changes at bolting (e.g. van der Graaff et al. 2006 DOI:10.1104/pp.106.079293, Parlitz et al. 2011 DOI:10.1016/j.jplph.2011.02.001, Law et al. 2018 DOI:10.1104/pp.18.00062, van der Graaff et al. 2006 DOI:10.1104/pp.106.079293, Hinckley and Brusslan 2020 DOI:10.1002/pld3.279). It would be interesting to add to the discussion a comparison between the results in *A. halleri* self-shading and sink-removal treatments and natural senescence and with the dark/shade-induced leaf senescence and developmental bolting-associated leaf senescence in *A. thaliana*.

Check the spelling of *A. halleri* throughout the manuscript

L170-171: In the text authors state that shading induced senescence, however in figure 3b, exposed leaves show earlier senescence than the shaded ones.

Reviewer #3 (Remarks to the Author):

In this manuscript, Yumoto et al. presents a comprehensive 4-year study backed by extensive datasets showing the leaf senescence during seasonal switch in the evergreen

perennial *Arabidopsis*. Their main conclusion is that a seasonal trend of leaf senescence is suspended during winter but become active during the growth and reproductive phases. Specifically, the authors track leaf phenotypes, including emergence, longevity, and growth rate, biweekly over the span of four years. The following analysis find a linkage between leaf phenotypes and seasonality, leading the authors to define the GS and OW cohorts (Fig. 1 and Fig. 2). The authors next focus on leaf senescence and employ transcriptome analysis (Fig. 3 and Fig. 4) to propose a model connecting plant senescence controls across three seasonal phases: growth, overwintering, and reproduction (Fig. 5). This is a pretty interesting work. I commend the authors for their long-term research on the perennial *Arabidopsis* and the large datasets provided, but I still have some points below that the authors should consider.

Major comments:

1. Lack of overall view of the transcriptome analysis

Much of the initial conclusion about the self-shading and sink-removal experiments rely on the transcriptome analysis with heat map showing the expression of few genes from each pathway. It seems important if the authors can show an overall view (big picture) of the two batch of the transcriptome analysis. What the authors can think about is reposition figure 5a prior to figures 3 and 4, followed by a deeper unbiased exploration of the DEGs data. Specifically, I'm interested in understanding the transcriptomic comparisons at week zero for GS and OW as shown in figure 3. Would senescence-related pathways be highlighted in a Gene Set Enrichment Analysis (GSEA) at this time? In addition, a deeper analysis to the distribution of the 308 senescence-related DEGs shown in figure 5a would be informative. It would also be helpful to know the number and extent of these DEGs associated with the eight pathways highlighted in figures 3 and 4.

2. Only nine genes analyzed in each senescence-related pathway

In figure 5a, the authors identified 308 senescence-related DEGs from the transcriptomic analysis of the self-shading and sink-removal experiments. Then eight senescence-related

pathways were analyzed for both experiments. However, the authors didn't clarify the reasons for choosing the nine genes in each pathway for further analysis. I would like to know how many genes in the 308 DEGs are belong to each pathway.

3. Removing new leaves for GS vs removing flowering stalks for OW

It is essential to recognize that flowering plants invest a significant amount of energy and resources into producing flowers and subsequently seeds. Maybe the most concern I have for this manuscript is that the authors designed the reproductive-sink-triggered senescence experiment with leaves removal in GS but flower stalks removal in OW, and then concluded with desynchronized leaf senescence in growth and synchronized leaf senescence in reproduction. Maybe the authors have reasons to perform sink-triggered senescence in this way, like no flower stalks in GS? Please clarify it in the text this if I misunderstood. But in my understanding, removing new leaves and flowering stalks are totally different to plants, particularly in senescence response.

Minor points:

1. Line 26 in the abstract, what means "secondary" controls? which one is the primary control? The whole sentence "The secondary controls of leaf senescence resulted in desynchronized and synchronized leaf senescence during growth and reproduction, respectively." is confusing to me. Please clarify it.

2. Figure 1. I believe the right three panels should be labeled as e, f, and g.

3. It is very interesting to see a biphasic growth rate in figure 1d. Do the authors have opinions why it is biphasic?

4. line 136-137, what does "while the age structure gradually developed into leaves up to 8 months" mean?

Response letter

We are grateful for comments on our manuscript, which help us to improve it significantly. In the following letter, the original comments are in italics and we have renumbered the comments (in bold). Our responses follow each numbered comment in plain text. We have prepared two types of Word files for the revised version, with and without the track change. The line numbers in our responses correspond to those in the file without track changes. Please see the track-change version to see where we made changes. This time we have used a professional English editing service by a native English-speaking editor, and we have also incorporated their suggested changes. We hope that the manuscript has become good enough to be published in Nature Communications, although we are willing to do any further editing necessary to improve our manuscript on any point.

Reply to the Reviewers' comments:

Comments to Authors from Editor in chief:

Thank you for reviewing our manuscript. We have revised the manuscript in accordance with the comments and our response to each comment is given below.

REVIEWER COMMENTS

Reviewer #1 and #2 (Remarks to the Author):

1. *By studying evergreen perennial plant *Arabidopsis halleri*, the authors identified factors that control leaf senescence in different seasons. Although the results are not that surprising, they are interesting, they fill a gap in the literature and improves our understanding on how senescence is coordinated. Although it is well known that regulation of senescence is very complex, this study illustrates well how complicated it is to make generalizations, as the developmental stage of the plant determines how it responds to environmental cues.*

The study was conducted in the field over four years and the authors measured exhaustively the growth and longevity of leaves and identified two distinct seasons: growth season and overwintering season, that were separated by photoperiod. Shading and sink-removal treatments had different effects on leaf longevity in different cohorts indicating that the secondary controls for senescence differed between seasons. Shading delayed leaf senescence in the growth season, but not in the overwintering season. Leaf

senescence at the end of winter season was on the other hand synchronized with flowering and the removal of sinks delayed leaf senescence more in overwintering season than in growth season. Overall, the leaf growth and longevity data have been thoroughly analysed, well presented and the results appear sound.

Thank you for reviewing our manuscript.

2. In addition, the authors studied gene expression in different treatments in the two leaf cohorts and noted that senescence-associated gene (SAG) expression was off in overwintering leaves. I see a lot of potential in the generated time-series gene expression data; however, its presentation could be improved.

We agreed with the comment that there is a lot of potential to improve the manuscript by adding further analyses on the time series gene expression data. We have extensively revised this aspect. Please see the responses to the following comments.

3. Methods are overall well described; however, I would recommend authors to better explain how the plants were sampled in the experiments. I do not really understand if the same plants sampled repeatedly during the time course over the season? And were the sampled plants included or excluded from the monitoring group for leaf longevity in the treatment plots? Since sink removal affected leaf longevity, especially in OW group, could leaf sampling affect leaf longevity as well? Were the shaded and exposed leaves sampled from the same plant individual?

We agree with the comment and have added new supplementary figures illustrating the sampling methods for the self-shading (**Supplementary Fig. 7**) and sink-removal (**Supplementary Fig. 8**) experiments. All questions about the sampling methods are clearly stated as 'notes' in the figure.

For those plants from which a leaf was removed for RNA sampling during the experiments, we used records of other remaining leaves for leaf longevity analyses. Leaf longevity records were not significantly different between plants with and without RNA sampling (data not shown), and sampling two and one leaves per plant once during the self-shading and sink-removal experiments, respectively, had little effect on leaf longevity analyses. Note that the removal of new leaves for the GS sink-removal treatment was done every week, whereas RNA sampling was done once per plant during the experiments.

We have also added sentences to explain the detailed sampling procedures in the Methods section (**L. 609-612, 626-628, 631-646, 659-662**).

4. Authors explained that in the sink-removal experiment, the control group was similarly disturbed by touching the plants. The effect of sink-removal could be partially due to the stress response induced by physical damage. Was there a control group for physical damage?

For the controls in the sink removal treatments, we handled the plants at the same intensity as the removal treatments by holding the flowering stem and touching the base of the stem with a pair of scissors. However, there was no physical damage to the control plants in the form of tissue injury. Therefore, it cannot be completely excluded that the effect of sink-removal is partly due to the stress response induced by physical damage. In the revised version, we have evaluated this possibility by examining wounding-responsive genes (added as a new **Supplementary Fig. 6**) and briefly discussed the points in the main text (**L. 321-330**).

5. Transcriptomics results are focused on the well-established set of senescence-associated genes (SAGs) and the expression patterns of some of them are visualised in the heatmaps in figures 3 and 4. Although this is acceptable; it overlooks the other affected processes beyond the well-established change in the SAG expression. Differential expression analysis showed that thousands of genes were affected by the treatments in each leaf cohort. Authors compared treatment and control groups separately in each time point and cohort. However, it remains unclear to me how the DE genes were defined as up- or down-regulated in a time-series.

I would recommend authors to compare the complete lists of DE genes, not just the SAGs, between the cohorts and different treatments. The list of DE genes affected in each condition and if they are shared between any of the treatments could be added as supplementary information.

In the revision, we clearly defined up- and down-regulated DEGs in a time series (**L. 182-184, 276-277**) and added **Fig. 3d-f** and **Fig. 4d-f** to show how many times the DEGs were detected in the time series for all and senescence-related DEGs (**L. 184-188, 192-200, 277-279 and 281-285**). For the comparisons between cohorts and treatments, we generated new Venn diagrams in **Fig. 5a, b** to show the number of genes shared between cohorts and treatments for all genes in addition to senescence-related genes (**Fig. 5e, f**) and described the results in the main text (**L. 333-344, 359-373**).

6 It would be interesting to see what are the shared 266 up- and 636 down-regulated genes in GS self-shading and OW sink-removal treatments or the shared 64 genes up-regulated in self-shading treatment in both cohorts as indicated in supplementary figure 5? These analyses could be easily combined with enrichment analyses for the identified gene sets that could provide useful information of what kind of biological processes are involved in senescence regulation in each treatment and season.

Following the comments, we performed GO enrichment analysis for unique and shared DEGs between GS self-shading and OW sink-removal in a revised **Fig. 5a-d**. We found interesting patterns of enriched GOs in both shared and unique DEGs for up- and down-regulated DEGs and included the results in the main text (**L. 345-358**).

7 What calls my attention is the high fold changes which are NOT significant especially in GS sink-removal group; this indicates high variation within the sample group. I would suggest that the authors to check the quality of their data before DE analyses and identify potential outliers. For example, clustering analysis or principal component analysis could be used to identify if there are outliers in the data.

We performed PCA analyses to show how each sample located in the PCA space (**Supplementary Fig. 5**). As noted in the comment, samples showed high variation between replicates in the GS sink-removal experiment (**Supplementary Fig. 5c**). Rather than having outliers, high variation between replicates was repeatedly detected at multiple time points of both treatments. As the variation in leaf longevity was high within the GS cohorts, we considered that the high transcriptomic variation was generated depending on the local environmental variation of leaves in the GS cohort. We have added this argument in the main text of the revised version (**L. 309-313**).

8 Another thing is that there are several DE genes in all the treatments at the time point 0. Were the samples collected before the treatments started or after during the same day? Was this initial difference at time 0 considered in the subsequent analyses of the gene expression data?

At time point 0, samples were collected one hour after treatments on the same day. In the Results section of the revised version (**L. 180, 272**), we have added a more detailed explanation of the time points of RNA sampling. Therefore, we treated the difference at time zero as part of the responses to treatments, i.e. DEGs.

9 *The treatments in GS and OW were performed two years apart, not during consecutive seasons. It would still be interesting to study what kind of processes are differently regulated between GS leaves and OW leaves in green leaves, during senescence process and in response to different treatments. This would provide some additional information on how OW leaves differ from GS leaves. Therefore, I believe that the study would benefit from additional analyses of the transcriptomics data that could perhaps shed more light into how senescence is induced or prevented during different seasons and in different treatments.*

Thanks to the comment, we checked our sampling design to see if we could compare intact GS and OW leaves for their transcriptomes. As the sink+ plants in the sink-removal experiments were intact plants and GS and OW samples shared four time points after leaf emergence, we added a new analysis comparing them and created **a new Fig. 6**. As transcriptomic differences in this set of comparisons should reflect not only senescence but also any other environmental differences between the growing and overwintering seasons, we compared the expression patterns of the senescence-related genes and added a new paragraph to explain our findings (**L. 375-420**).

10 *I would also recommend authors to extend the discussion of their results in line with existing literature. For example, in an evergreen perennial tea tree (*Camellia sinensis*), SAGs were not up-regulated in the winter leaves, instead the genes related to stress tolerance were highly expressed (Paul et al. 2014, DOI: 10.1038/srep0593). In addition, gene expression changes during dark-induced and developmental leaf senescence have been extensively studied with *Arabidopsis thaliana* and it has been shown that SAG expression changes at bolting (e.g. van der Graaff et al. 2006 DOI:10.1104/pp.106.079293, Parlitz et al. 2011 DOI:10.1016/j.jplph.2011.02.001, Law et al. 2018 DOI:10.1104/pp.18.00062, van der Graaff et al. 2006 DOI:10.1104/pp.106.079293, Hinckley and Brusslan 2020 DOI:10.1002/pld3.279). It would be interesting to add to the discussion a comparison between the results in *A. halleri* self-shading and sink-removal treatments and natural senescence and with the dark/shade-induced leaf senescence and developmental bolting-associated leaf senescence in *A. thaliana*.*

Thank you for suggesting related literature. We have included the listed literature in the main text and added related discussion in the revised version.

Paul et al. 2014 (**L. 451 - 456** in Discussion).

van der Graaff et al. 2006 (**L. 212-214** in Results).

Parlitz et al. 2011 (L. 466-469 in Discussion).

Law et al. 2018 (L. 469-471 in Discussion).

Hinckley and Brusslan 2020 (L. 70-71 in Introduction, L. 491-493 in Discussion).

11 *Check the spelling of A. halleri throughout the manuscript*

We checked and corrected spelling throughout the manuscript.

12 *L170-171: In the text authors state that shading induced senescence, however in figure 3b, exposed leaves show earlier senescence than the shaded ones.*

This was an error in the labelling of the figure, and the shading induced senescence. We apologise for the error. We have corrected the mislabeling (**Fig. 3b,c**).

Reviewer #3 (Remarks to the Author):

13 *In this manuscript, Yumoto et al. presents a comprehensive 4-year study backed by extensive datasets showing the leaf senescence during seasonal switch in the evergreen perennial Arabidopsis. Their main conclusion is that a seasonal trend of leaf senescence is suspended during winter but become active during the growth and reproductive phases. Specifically, the authors track leaf phenotypes, including emergence, longevity, and growth rate, biweekly over the span of four years. The following analysis find a linkage between leaf phenotypes and seasonality, leading the authors to define the GS and OW cohorts (Fig. 1 and Fig. 2). The authors next focus on leaf senescence and employ transcriptome analysis Fig. 3 and Fig. 4) to propose a model connecting plant senescence controls across three seasonal phases: growth, overwintering, and reproduction (Fig. 5). This is an pretty interesting work. I commend the authors for their long-term research on the perennial Arabidopsis and the large datasets provided, but I still have some points below that the authors should consider.*

Thank you for reviewing our manuscript. We have extensively revised the manuscript in response to the reviewer's comments, and please see our response to each comment.

14 *Major comments:*

1. Lack of overall view of the transcriptome analysis

Much of the initial conclusion about the self-shading and sink-removal experiments rely on the transcriptome analysis with heat map showing the expression of few genes from each pathway. It seems important if the authors can show an overall view (big picture) of the two batch of the transcriptome analysis. What the authors can think about is reposition figure 5a prior to figures 3 and 4, followed by a deeper unbiased exploration of the DEGs data.

Following the comments, we have expanded the analyses that were partially shown in the previous Fig. 5a. In the revised version, we compared GS self-shading and OW sink-removal treatments for all DEGs (**Fig. 5a-d, L. 332-358**) and senescence-related genes (**Fig. 5e,f, L. 359-373**) to show an overall view (how two transcriptome batches overlap and how they are unique by performing the GO enrichment analyses for the common and unique DEGs). We have not placed the information in Fig. 5 before Figs. 3 and 4 because the comparisons between self-shading and sink-removal experiments (Fig. 5) require some of the information presented in Figs. 3 and 4, which is difficult to separate from other information in these Figs. Instead, by adding new graphs, **d-e, to Figs. 3 and 4**, we have explained how many DEGs were detected for both all genes and senescence-related genes (**corresponding explanations in L. 184-188, 192-200, 277-286**).

15 Specifically, I'm interested in understanding the transcriptomic comparisons at week zero for GS and OW as shown in figure 3. Would senescence-related pathways be highlighted in a Gene Set Enrichment Analysis (GSEA) at this time?

Firstly, week zero represents samples collected one hour after treatments on the day treatments started. We have added a more detailed explanation of the times of RNA sampling in the results sections of the revised version (**L. 178-182, 268-272**). Instead, we further compared the OW and GS intact cohorts using the sink+ treatments dataset in the sink removal experiments (**Fig. 6, L. 375-420**). We used gene ontology enrichment analyses as part of the GSEA (**Supplementary Data 8**). Senescence-related GO was enriched in GS-upregulated DEGs and we described the results briefly in the main text (**L. 384-388**). Although KEGG pathway analyses showed enrichment of 'Photosynthesis' for GS upregulated DEGs, it can be explained by higher photosynthetic activity during growth seasons. Because other senescence related pathway (porphyrin and chlorophyll metabolism) was not significant, we decided exclude the results of KEGG enrichment analyses due to space limitation. See also the response to **Comment 9**.

16 *In addition, a deeper analysis to the distribution of the 308 senescence-related DEGs shown in figure 5a would be informative. It would also be helpful to know the number and extent of these DEGs associated with the eight pathways highlighted in figures 3 and 4.*

The number of senescence-related genes was updated from 308 to 432 in the revised version (**Supplementary Table 2**). We added a new graph **g** in **Figs. 3 and 4** (L. 210-216, 287-291) to show the distribution of senescence-related DEGs in eight categories (+ others). We updated **former Fig. 5a** to **new Fig. 5 e, f**, and listed diagrams showing distribution of senescence-related DEGs across categories for unique DEGs to either GS self-shading or OW sink removal experiment and common DEGs separately (L. 359-373).

17 *2. Only nine genes analyzed in each senescence-related pathway*

In figure 5a, the authors identified 308 senescence-related DEGs from the transcriptomic analysis of the self-shading and sink-removal experiments. Then eight senescence-related pathways were analyzed for both experiments. However, the authors didn't clarify the reasons for choosing the nine genes in each pathway for further analysis. I would like to know how many genes in the 308 DEGs are belong to each pathway.

We have presented a number of analyses not only for senescence genes but also for other genes. In **Figs. 3** (self-shading experiments) and **4** (sink-removal experiments), we added new plots (**d, e, f, g**) to show how many total DEGs were up- and down-regulated in senescence-promoting treatments, how many of them were senescence-related, and how they were categorised into different senescence pathways. This clarified how selected genes on the heatmaps represented overall senescence responses.

17 *3. Removing new leaves for GS vs removing flowering stalks for OW*

It is essential to recognize that flowering plants invest a significant amount of energy and resources into producing flowers and subsequently seeds. Maybe the most concern I have for this manuscript is that the authors designed the reproductive-sink-triggered senescence experiment with leaves removal in GS but flower stalks removal in OW, and then concluded with desynchronized leaf senescence in growth and synchronized leaf senescence in reproduction. Maybe the authors have reasons to perform sink-triggered senescence in this way, like no flower stalks in GS? Please clarify it in the text this if I misunderstood. But in my understanding, removing new leaves and flowering stalks are totally different to plants,

particularly in senescence response.

As the referee pointed out, for the sink removal experiments we removed new leaves in the GS experiment because there were no flowering stems during the growing season. In addition, leaf production is much higher during the growing season, implying that new leaves are the strongest demand for the GS cohort. We have clarified these points in the revised version to make it clear how we chose the experimental treatments (**L. 261-264**).

18 *Minor points:*

1. Line 26 in the abstract, what means “secondary” controls? which one is the primary control? The whole sentence “The secondary controls of leaf senescence resulted in desynchronized and synchronized leaf senescence during growth and reproduction, respectively.” is confusing to me. Please clarify it.

We have edited sentences in the abstract to make it clear what the primary and secondary controls are (**L. 21-27**).

19 *2. Figure 1. I believe the right three panels should be labeled as e, f, and g.*

The comment is correct. We have corrected it in the revised version.

20 *3. It is very interesting to see a biphasic growth rate in figure 1d. Do the authors have opinions why it is biphasic?*

As the period of slower growth of the cohorts extended from June to September, the high temperature regime could explain the growth suppression in summer. We have added this argument in the revised version (**L. 106-110**).

21 *4. line 136-137, what does “while the age structure gradually developed into leaves up to 8 months” mean?*

We have revised the sentence to clarify its meaning (**L. 137-140**).

The end of the letter.

REVIEWERS' COMMENTS

Reviewers #1 and #2 (Remarks to the Author):

I have read the revised version of the Ms and think the authors have addressed my concerns in an appropriate way, hence I recommend that the ms should be accepted. I should say though that is language is sometimes not very smooth. It does not make the ms hard to understand or reduce the scientific value, it is more a matter of taste. If it gets accepted maybe an editor with English as mother tongue could suggest some changes.

Reviewer #3 (Remarks to the Author):

It was a pleasure to read the revised version of the manuscript. The authors have addressed all my comments, and the manuscript has improved. In particular, the expansion of figures 3 and 4, along with the two additional figures, enhances the comprehensive comparison of DEGs and biological processes between OW and GS.

Responses to reviewers' comments

We listed original comments in italics and our responses in plain text.

Reviewer #2 (Remarks to the Author):

I have read the revised version of the Ms and think the authors have addressed my concerns in an appropriate way, hence I recommend that the ms should be accepted. I should say though that is language is sometimes not very smooth. It does not make the ms hard to understand or reduce the scientific value, it is more a matter of taste. If it gets accepted maybe an editor with English as mother tongue could suggest some changes.

Thank you for evaluating our manuscript. The revised version of our manuscript was sent to the professional English editing service, and we revised it following their suggestions.

Reviewer #3 (Remarks to the Author):

It was a pleasure to read the revised version of the manuscript. The authors have addressed all my comments, and the manuscript has improved. In particular, the expansion of figures 3 and 4, along with the two additional figures, enhances the comprehensive comparison of DEGs and biological processes between OW and GS.

Thank you for evaluating our manuscript. Thanks for the reviewers' comments, our manuscript has been improved a lot.